# PARCO: Parallel AutoRegressive Models for Multi-Agent Combinatorial Optimization

**Federico Berto**[*,1,2,3], **Chuanbo Hua**[*,1,2], **Laurin Luttmann**[*,4], **Jiwoo Son**[2], **Junyoung Park**[1],
**Kyuree Ahn**[2], **Changhyun Kwon**[1,2], **Lin Xie**[5], **Jinkyoo Park**[1,2]

[1]KAIST  [2]Omelet  [3]Radical Numerics  [4]Leuphana University
[5]Brandenburg University of Technology  AI4CO[‡]

## Abstract

Combinatorial optimization problems involving multiple agents are notoriously challenging due to their NP-hard nature and the necessity for effective agent coordination. Despite advancements in learning-based methods, existing approaches often face critical limitations, including suboptimal agent coordination, poor generalization, and high computational latency. To address these issues, we propose PARCO (Parallel AutoRegressive Combinatorial Optimization), a general reinforcement learning framework designed to construct high-quality solutions for multi-agent combinatorial tasks efficiently. To this end, PARCO integrates three key novel components: (1) transformer-based communication layers to enable effective agent collaboration during parallel solution construction, (2) a multiple pointer mechanism for low-latency, parallel agent decision-making, and (3) priority-based conflict handlers to resolve decision conflicts via learned priorities. We evaluate PARCO in multi-agent vehicle routing and scheduling problems, where our approach outperforms state-of-the-art learning methods, demonstrating strong generalization ability and remarkable computational efficiency. We make our source code publicly available to foster future research: https://github.com/ai4co/parco.

## 1 Introduction

Combinatorial optimization (CO) problems involve determining an optimal sequence of actions in discrete spaces with several crucial domains, including logistics and supply chain management [91]. Many practical CO problems require a solution to be constructed by coordinating multiple distinct entities (i.e., agents), each with unique characteristics. We call such problems *multi-agent CO*. This class of problems naturally arises in real-world applications such as coordinated vehicle routing for disaster management [60], manufacturing [37] and last-mile delivery optimization [1], where heterogeneous agents must operate under complex constraints.

CO problems are notoriously hard to solve and cannot generally be solved optimally in polynomial time due to their NP-hardness [36, 79]. While traditional methods such as exact and heuristic methods have

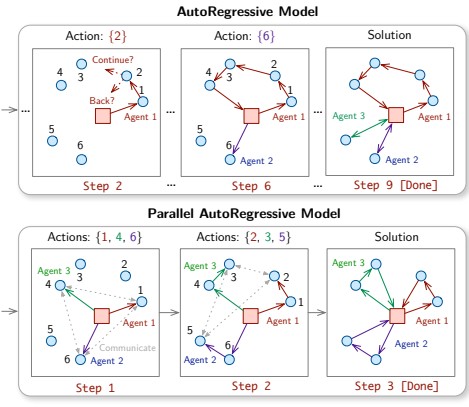

Figure 1: PARCO generates better solutions with higher efficiency through parallel decision making.

---

[*]Equal contributions.

[‡]Authors are members of the AI4CO open research community.

39th Conference on Neural Information Processing Systems (NeurIPS 2025).

been developed to solve a variety of problems [51, 25], these approaches often concentrate on single-agent scenarios and require long execution times. Moreover, multi-agent CO poses additional challenges including additional constraints and different optimization objectives, such as minimizing lateness or the makespan among agents [82, 72].

Recently, Neural Combinatorial Optimization (NCO) has emerged as a powerful alternative by learning efficient neural solvers [4]. In particular, Reinforcement Learning (RL) has shown promise due to its ability to learn directly from interactions with CO problems instead of relying on costly labeled datasets in the shape of optimal solutions, and could even outperform traditional approaches by automatically discovering better (neural) heuristics [3, 49, 40]. Among NCO methods, Autoregressive (AR) models—constructing solutions step-by-step—have garnered attention for their ability to generate solutions for a variety of problems with hard constraints [47, 50, 7]. This capability is crucial for addressing complex problems with multiple constraints, such as heterogeneous capacity [2] or precedences in pickup and delivery [77] and machine scheduling problems [101].

However, despite their promising results in single-agent settings, AR-CO methods pose two challenges that hinder their practical adoption in solving complex multi-agent CO problems. First, existing AR methods for multi-agent CO suffer from poor coordination, resulting in unsatisfactory solution quality and poor generalization across varying problem sizes and agent configurations [108, 84, 104, 63]. Moreover, AR sequence generation is associated with high latency due to each single action (or "token") depending on each previous one, akin to large language models [52, 21]. This issue becomes increasingly pronounced when dealing with large problem instances involving numerous agents.

This paper introduces PARCO (Parallel AutoRegressive Combinatorial Optimization), a novel learning framework to address multi-agent combinatorial problems effectively via parallel solution construction as illustrated in Fig. 1. We design specialized transformer-based Communication Layers to enhance coordination among agents during decision-making, enabling collaborative behaviors. Our model leverages a Multiple Pointer Mechanism to efficiently generate actions for various agents simultaneously at each solution construction step. Priority-based Conflict Handlers ensure feasibility and resolve potential conflicts among agents' decisions based on learned priorities.

We summarize PARCO's contributions as follows:

- We propose a general Parallel Autoregressive framework for solution construction in multi-agent CO.
- We introduce Communication Layers to enhance agent coordination at each parallel construction step.
- We design a Multiple Pointer Mechanism that reduces latency by efficiently decoding solutions in parallel.
- We enhance solution quality with Priority-based Conflict Handlers that tie break with learned priorities.
- We evaluate PARCO on multi-agent vehicle routing and scheduling, where we outperform state-of-the-art learning methods in solution quality, generalization, and efficiency.

## 2 Related Work

**Neural Combinatorial Optimization**    Recent advancements in Neural Combinatorial Optimization (NCO) have shown promising end-to-end solutions for combinatorial optimization problems [4, 94]. NCO has led to the development of a variety of methods for diverse problems, including the incorporation of problem-specific biases [35, 64, 44, 41, 83, 20, 31], bi-level solution pipelines [56, 96, 105], learning-guided search [95, 45, 85, 57, 58, 59, 42], improvement methods [26, 69, 70, 28], effective training algorithms [43, 23, 14, 13, 80, 65, 103], downstream applications [11, 106, 71, 66, 93], and the recent development of end-to-end foundation models [61, 107, 15, 6, 34, 53, 22]. Among such methods, RL-based end-to-end autoregressive (AR) models present several advantages, including eliminating the need for labeled solutions, reducing reliance on handcrafted heuristics, and achieving high efficiency in generating high-quality solutions [47, 49, 50, 5, 8].

**Multi-Agent AR Methods for CO**    While the seminal works in AR-CO methods of Vinyals et al. [89], Kool et al. [47], Kwon et al. [49, 50] propose models that can be used in loose multi-agent settings, such methods cannot be employed directly to model heterogeneous agents with different

attributes and constraints. Building on Kool et al. [47], Son et al. [84] and Zheng et al. [104] introduce attention-based policies for multi-agent min-max routing. These models adopt a sequential AR construction strategy, solving for one agent at a time and switching agents only after completing a single-agent solution. While this approach outperforms decentralized methods [10, 76] it remains inherently sequential. In contrast, other multi-agent AR methods determine a location for every agent in each solution construction step, but the different agents select their actions sequentially in either random [102] or learned order [18, 55, 63], making these models still suffer from high generation latency as well as missing inter-agent communication and coordination. Zong et al. [108] propose a multi-agent pickup and delivery model with parallel decoding, using distinct decoders for each agent. However, this approach exhibits limited generalizability due to inflexible fixed decoders for specific agents, lacking a powerful communication mechanism, and conflict resolution handled naively by assigning random precedence to agents, restricting the robustness of the model.

PARCO addresses the shortcomings of previous works by leveraging parallel solution construction for any number of agents efficiently with a Multiple Pointer Mechanism, enhancing coordination via Communication Layers, and solving conflicts in a principled manner via Priority-based Conflict handlers. Finally, unlike previous works, PARCO is a general framework tackling multi-agent CO without restricting to a single class of problems.

## 3 Preliminaries

### 3.1 Markov Decision Processes

Multi-agent CO problems can be formulated as Markov Decision Processes (MDPs) and solved autoregressively using RL [74]. In this framework, a solution $\boldsymbol{a}$ to a CO problem instance $\boldsymbol{x}$ is represented as a sequence of actions. Actions $a_t$ are selected sequentially from the action space $\mathcal{A}$ of size $N$ based on the current state $s_t \in \mathcal{S}$, which encodes the problem's configuration at step $t$. In multi-agent problems, at each step one agent $m \in \mathcal{M} = \{1, \dots, M\}$ selects an action $a_t^m$ according to a policy $\pi_\theta$, usually represented by a $\theta$-parametrized neural network, mapping states to actions. Agents are selected either by some predefined precedence rule as in the sequential planning of Son et al. [84], where agent solutions are constructed one after another, or by the policy itself, in which case $\pi_\theta : \mathcal{S} \to \mathcal{A} \times \mathcal{M}$. Given the agent and its corresponding action, the problem then transitions from state $s_t$ to state $s_{t+1}$ according to a transition function $\tau : \mathcal{S} \times \mathcal{A} \times \mathcal{M} \to \mathcal{S}$. This process reaches the terminal state once it has generated a feasible solution $\boldsymbol{a} = (a_1, ..., a_T)$ for the problem instance $\boldsymbol{x}$ in $T$ construction steps. The (sparse) reward $R(\boldsymbol{a}, \boldsymbol{x})$ is usually obtained only in the terminal state and takes the form of the negative of the cost function of the respective CO problem.

### 3.2 AR Models for CO

Given the sequential nature of MDPs, autoregressive (AR) models pose a natural choice for the policy $\pi_\theta$. AR methods construct a viable solution by sequentially generating actions based on the current state and previously selected actions. Without loss of generality, the process can be represented in an encoder-decoder framework as:

$$p_\theta(\boldsymbol{a}|\boldsymbol{x}) \triangleq \prod_{t=1}^{T} g_\theta(a_t|\boldsymbol{a}_{<t}, \boldsymbol{h}) \tag{1}$$

where $\boldsymbol{a}_{<t} = (a_1, \dots, a_{t-1})$ is the sequence of actions taken prior to $t$ and $\boldsymbol{h} = f_\theta(\boldsymbol{x})$ is an encoding of the problem instance $\boldsymbol{x}$ obtained via the encoder network $f$. The decoder $g_\theta$ then autoregressively generates the sequence of actions, conditioned on $\boldsymbol{h}$ and the previously generated actions. The parameters $\theta$ encompass both the encoder and decoder components, which together define the policy as the mechanism for producing the joint distribution $p_\theta(\boldsymbol{a}|\boldsymbol{x})$. Thus, the RL objective becomes finding the optimal set of parameters $\theta^*$ that maximizes the reward function $R$ [48, 49, 43].

## 4 Methodology

We now outline the general structure of PARCO as shown in Fig. 2. First, we formally define parallel multi-agent MDPs for CO (§ 4.1), which we use as a basis to derive our overall parallel autoregressive approach (§ 4.2). We then describe in detail the components of our model: Multi-Agent Encoder

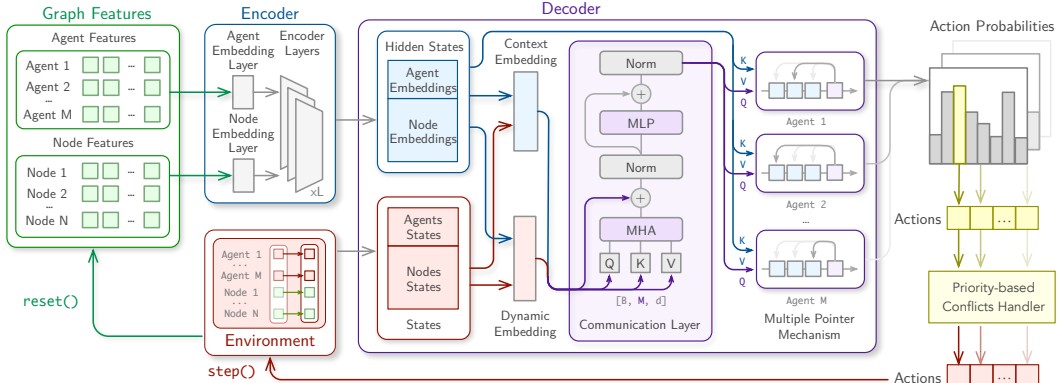

Figure 2: Overview of PARCO. Our model encodes multi-agent CO problems into separate agent and node embeddings. Communication Layers allow for coordination among agents during decoding, which enhances solution quality. Actions are decoded efficiently autoregressively in parallel through a Multiple Pointer Mechanism enhanced by a Priority-based Conflict Handler.

(§ 4.3), Communication Layers (§ 4.4), Decoder with Multiple Pointer Mechanism (§ 4.5) and Conflict Handlers (§ 4.6). Finally, we outline the training scheme (§ 4.7).

## 4.1 Cooperative Multi-Agent MDPs

We reformulate the MDPs of § 3.1 as cooperative multi-agent MDPs [9], often termed fully cooperative Markov games [73], by selecting multiple actions from a joint action space simultaneously to enhance efficiency and coordination [98, 68]. At each step $t$, $M$ agents select actions $\boldsymbol{a}_t = (a_t^1, ..., a_t^M)$ according to a policy $\pi_\theta : \mathcal{S} \rightarrow \mathcal{A}_1 \times ... \times \mathcal{A}_M$, which maps the state space $\mathcal{S}$ to the joint action space of agents. A conflict handling function $\psi : (\mathcal{A}_1 \times ... \times \mathcal{A}_M) \rightarrow (\mathcal{A}_1 \times ... \times \mathcal{A}_M)$ ensures action compatibility by allowing only one agent to execute when multiple agents select mutually conflicting actions (e.g., the same customer location), and assigning a fallback action (e.g., staying at current position) to the others. Given the resolved agent actions $\tilde{\boldsymbol{a}}_t = \psi(\boldsymbol{a}_t)$, the state of the problem $s_t$ progresses to $s_{t+1}$ according to the transition function $\tau : \mathcal{S} \times (\mathcal{A}_1 \times ... \times \mathcal{A}_M) \rightarrow \mathcal{S}$. The agents receive a shared reward $R(\boldsymbol{a}, \boldsymbol{x})$, with $\boldsymbol{a} = (\tilde{\boldsymbol{a}}_1, ..., \tilde{\boldsymbol{a}}_T)$ the sequence of joint agent actions.

In the following, we refer to the entities corresponding to actions in the MDP formulation of a CO problem (e.g., customer locations in VRPs) as *nodes*, following the convention of Kool et al. [47] and Kwon et al. [49].

## 4.2 Parallel AR Models for CO

Motivated by the nature of multi-agent MDPs, PARCO introduces a Parallel AR model for the policy $\pi_\theta$. PARCO constructs feasible solutions by simultaneously generating multiple agent actions based on the current state. We formulate the solution generation process in an encoder-decoder framework similarly to Eq. (1):

$$p_\theta(\boldsymbol{a}|\boldsymbol{x}) \triangleq \prod_{t=1}^{T} \psi \left( \prod_{m=1}^{M} g_\theta(a_t^m | \boldsymbol{a}_{<t}, \boldsymbol{h}) \right) \tag{2}$$

where $\boldsymbol{h} = f_\theta(\boldsymbol{x})$ is the encoding of problem instance $\boldsymbol{x}$ via encoder network $f_\theta$, and decoder $g_\theta$ implements the policy $\pi_\theta$ to autoregressively generate actions efficiently for all agents in parallel. At each step $t$, the decoder outputs joint actions $\boldsymbol{a}_t = (a_t^1, ..., a_t^M)$ for all $M$ agents with $a_t^m$ being the sampled action of agent $m$. The conflict resolution function $\psi$ ensures action compatibility by allowing only one agent to execute when multiple agents select mutually conflicting actions, resulting in resolved actions $\tilde{\boldsymbol{a}}_t = \psi(\boldsymbol{a}_t)$. $p_\theta$ is thus a solver that maps $\boldsymbol{x}$ to a solution $\boldsymbol{a} = (\tilde{\boldsymbol{a}}_1, ..., \tilde{\boldsymbol{a}}_T)$.

A benefit of our parallel formulation is that the total number of construction steps $T$ can be substantially lower compared to purely AR methods. While the latter require $\sum_{m=1}^{M} T_m$ total actions to construct a solution with $T_m$ being the number of steps required by agent $m$ to finish its task,

PARCO's Parallel AR needs only $\max_m T_m$ steps as agents effectively divide the solution space and act concurrently. This leads to a faster solution construction as illustrated in Figs. 1 and 6 and significantly reduced training times.

## 4.3 Multi-Agent Encoder

The multi-agent encoder $f_\theta$ transforms an input instance $\boldsymbol{x}$ into a hidden representation $\boldsymbol{h}$. In PARCO, we explicitly model agents and employ separate agent and node embedding layers similar to Son et al. [84] to project agents and nodes into the same embedding space.

The agent embedding layer projects $k_a$ agent features – such as vehicle locations and capacities (routing) or machine characteristics (scheduling) – into a $d$-dimensional space using a linear projection $\mathbf{W}_a \in \mathbb{R}^{k_a \times d}$. Let $\boldsymbol{h}_a^{(0)} = \boldsymbol{x}_a \mathbf{W}_a$ denote the initial agent embeddings, where $\boldsymbol{x}_a \in \mathbb{R}^{M \times k_a}$ denotes the matrix of agent features. Similarly, the node embedding layer projects $k_n$ node features – such as customer demands in vehicle routing or job durations in scheduling problems – into the same $d$-dimensional space using a linear projection $\mathbf{W}_n \in \mathbb{R}^{k_n \times d}$. The initial embeddings of the nodes are defined as $\boldsymbol{h}_n^{(0)} = \boldsymbol{x}_n \mathbf{W}_n$ where $\boldsymbol{x}_{n_i} \in \mathbb{R}^{N \times k_n}$ represents the node feature matrix.

Depending on the problem structure, the initial agent and node embeddings might either be concatenated as $\boldsymbol{h}^{(0)} = \text{Concat}(\boldsymbol{h}_a^{(0)}, \boldsymbol{h}_n^{(0)})$ and passed through $L$ transformer blocks, consisting of multi-head self-attention MHA$(\boldsymbol{h}^{(0)}, \boldsymbol{h}^{(0)}, \boldsymbol{h}^{(0)})$ and multi-layer perceptrons (MLPs) as defined in Vaswani et al. [88]. Or, agent and node embeddings are used separately as query and keys/values, respectively, in a cross-attention mechanism MHA$(\boldsymbol{h}_a^{(0)}, \boldsymbol{h}_n^{(0)}, \boldsymbol{h}_n^{(0)})$ akin to MatNet [50]. The final embeddings $\boldsymbol{h} = \{\boldsymbol{h}_a, \boldsymbol{h}_n\}$ emitted by the last encoder layer contain processed representations of both agents and nodes that capture their interactions as well as the overall problem structure.

## 4.4 Communication Layers

At each step $t$ of the decoding process of Eq. (2), given the encoded representations $\boldsymbol{h}$, we construct dynamic agent queries that capture the current state of both agents and the environment. For each agent $m$, we form a context embedding $\boldsymbol{d}_m = \text{Concat}(\boldsymbol{h}_{a^m}, \boldsymbol{h}_{\delta_t^m}, \boldsymbol{h}_e)$ consisting of the following components: (1) the (static) embedding of the agent $\boldsymbol{h}_{a^m}$; (2) a projection $\boldsymbol{h}_{\delta_t^m} = \delta_t^m \mathbf{W}_\delta \in \mathbb{R}^d$ of the agent's dynamics $\delta_t^m$ like its current location and capacity (routing) or the time until the agent becomes idle (scheduling); (3) a projection $\boldsymbol{h}_e = e_t \mathbf{W}_\delta \in \mathbb{R}^d$ of the dynamic environment features $e_t$ that encode the current problem's state. These dynamic embeddings are then projected into query vectors $\boldsymbol{q}_m = \boldsymbol{d}_m \mathbf{W}_q$ where $\mathbf{W}_q \in \mathbb{R}^{3d \times d}$ is a learnable projection matrix.

The resulting queries $\boldsymbol{q} = [\boldsymbol{q}_1, \dots, \boldsymbol{q}_M] \in \mathbb{R}^{M \times d}$ are then processed through communication layers comprising multi-head self-attention followed by an MLP:

$$\boldsymbol{q}' = \text{Norm}(\text{MHA}(\boldsymbol{q}, \boldsymbol{q}, \boldsymbol{q}) + \boldsymbol{q}) \tag{3}$$

$$\boldsymbol{q} = \text{Norm}(\text{MLP}((\boldsymbol{q}')) + \boldsymbol{q}') \tag{4}$$

where Norm denotes a normalization layer [33, 100]. These layers are inherently agent-count agnostic, allowing PARCO to handle arbitrary numbers of agents, making it more flexible and generalizable across different problems. Communication Layers allow agents to coordinate their actions by attending to both the problem structure and other agents' states while maintaining efficiency through parallel processing.

## 4.5 Decoder with Multiple Pointer Mechanism

PARCO's decoder improves the AR pointer mechanism [89, 47] – originally designed for single-agent scenarios and recently applied to sequential multi-agent planning with a single agent at a time [84, 104] – to handle multiple agents operating in parallel via a Multiple Pointer Mechanism.

Starting with the processed agent queries $\boldsymbol{q}$ that underwent communication, we first compute agent-specific representations through masked cross MHA:

$$\boldsymbol{q}' = \text{MHA}(\boldsymbol{q}, \ \boldsymbol{h}_n + \xi_t \mathbf{W}_\xi^K, \ \boldsymbol{h}_n + \xi_t \mathbf{W}_\xi^V; \ \mathbf{M}_t) \tag{5}$$

where $\xi_t \in \mathbb{R}^{N \times k_\xi}$ are dynamic node features which are projected via $\mathbf{W}_\xi^K, \mathbf{W}_\xi^V \in \mathbb{R}^{k_\xi \times d}$ for keys and values of the MHA, respectively. Further, $\mathbf{M}_t \in \mathbb{R}^{M \times N}$ is the current action mask at step $t$, avoiding agent representations to attend to infeasible actions.

We then obtain a joint logit space $\boldsymbol{u}$ across all agents:

$$\boldsymbol{u} = \beta \cdot \tanh\left(\frac{\boldsymbol{q}'(\boldsymbol{h}_n \mathbf{W}^L + \xi_t \mathbf{W}_\xi^L)^\top}{\sqrt{d}}\right) \tag{6}$$

with learnable parameters $\mathbf{W}^L \in \mathbb{R}^{d \times d}$, $\mathbf{W}_\xi^L \in \mathbb{R}^{k_\xi \times d}$ and $\beta$ is a scale parameter, set to 10 following Bello et al. [3] to enhance exploration. The output logits $\boldsymbol{u} \in \mathbb{R}^{M \times N}$ are masked by setting infeasible actions given mask $\mathbf{M}_t$ to $-\infty$. The joint probability distribution over all agent actions becomes:

$$p(\boldsymbol{a}_t | \boldsymbol{a}_{<t}, \boldsymbol{h}) = \prod_{m=1}^{M} \frac{\exp(\boldsymbol{u}_{m,a_t^m})}{\sum_{j=1}^{N} \exp(\boldsymbol{u}_{m,j})} \tag{7}$$

where $\boldsymbol{a}_t = (a_t^1, \ldots, a_t^M)$ represents the joint action across all agents at step $t$.

## 4.6 Conflict Handlers

When sampling from the probability distribution $\boldsymbol{p}$ generated by the Multiple Pointer Mechanism, multiple agents may select the same action simultaneously, which can result in an infeasible solution in several CO problems – for instance, in vehicle routing problems, usually only one agent is allowed to visit a customer node – and it becomes essential how to deal with such a situation effectively. Conflict handling (i.e., tie-breaking) can be achieved by allowing a single agent among a number of agents that are in conflict to continue with its action while others revert to fallback actions, e.g., staying in their current position. A simple approach introduced by Zong et al. [108] consists of randomly selecting an agent to perform the new action. However, this can be suboptimal since it excludes inductive biases that can be leveraged, such as learned representations.

---

**Algorithm 1** Priority-based Conflict Handler

---

**Require:** Actions $\boldsymbol{a} \in \mathbb{N}^M$, Priorities $\boldsymbol{p} \in \mathbb{R}^M$, Fallback actions $\boldsymbol{r} \in \mathbb{R}^M$
**Ensure:** Resolved Actions $\boldsymbol{a}' \in \mathbb{N}^M$
 1: $\sigma \leftarrow \text{argsort}(\boldsymbol{p}, \text{descending} = \text{True})$       // Sort indices based on priorities in descending order
 2: $\hat{\boldsymbol{a}} \leftarrow \boldsymbol{a}[\sigma]$       // Reorder actions according to priority
 3: $C \leftarrow \mathbf{0}^M$       // Initialize conflict mask
 4: **for** $i = 2$ to $M$ **do**       // Check for conflicts in reordered actions
 5:      **if** $\hat{a}_i \in \{\hat{a}_1, \ldots, \hat{a}_{i-1}\}$ **then**
 6:         $C_i \leftarrow 1$       // $C_i = 1$ indicates a conflict for index $i$
 7:      **end if**
 8: **end for**
 9: $\hat{\boldsymbol{a}} \leftarrow (1 - C) \odot \hat{\boldsymbol{a}} + C \odot \boldsymbol{r}$       // Resolve conflicts by assigning fallback actions
10: $\boldsymbol{a}' \leftarrow \hat{\boldsymbol{a}}[\sigma^{-1}]$       // Reorder resolved actions back to original order

---

In PARCO, we propose Priority-based Conflict Handlers that leverage *priorities* as a tie-breaking rule. Such priorities can be based on heuristics – such as giving priority to agents close to completion or whose action results in the smallest immediate cost – or on learned priorities. In the latter case, the model output probability values of the selected actions $p(\boldsymbol{a}_t)$ serve as an indicator for prioritizing certain agents: the higher their value, the higher the priority learned by the model to have those agents win the tie-break.

Algorithm 1 shows our efficient vectorized implementation of the Priority-based Conflict Handler algorithm. In practice, we augment the conflict handler $\psi$ from § 4.2 with priorities $\boldsymbol{p} := p(\boldsymbol{a}_t)$ and fallback actions $\boldsymbol{r}$, i.e. $\psi(\cdot) := \psi(\boldsymbol{a}_t, \boldsymbol{p}, \boldsymbol{r})$. Fallback actions in PARCO correspond to "do nothing" operations – maintaining an agent's current position (routing) or keeping a machine idle (scheduling) – which may result in slightly more solution construction steps but do not affect the final solution $\boldsymbol{a}$. This approach effectively handles conflicts by allowing the affected agents to reconsider their choices given the actions of (preceding) agents in the next decoding step.

## 4.7 Training Scheme

PARCO is a centralized multi-agent decision-making framework and can thus be trained by RL algorithms proposed in the single-agent NCO literature. We train PARCO by employing the REINFORCE gradient estimator [92] with a shared baseline as outlined by Kwon et al. [49] and Kim et al. [43]:

$$\nabla_\theta \mathcal{L} \approx \frac{1}{B \cdot S} \sum_{i=1}^{B} \sum_{j=1}^{S} G_{ij} \nabla_\theta \log p_\theta(\boldsymbol{a}_{ij}|\boldsymbol{x}_i) \tag{8}$$

where $B$ is the batch size, $S$ the number of shared baseline samples, and $G_{ij} = R(\boldsymbol{a}_{ij}, \boldsymbol{x}_i) - b_i^{\text{shared}}(\boldsymbol{x}_i)$ is the advantage of a solution $\boldsymbol{a}_{ij}$ compared to the shared baseline $b_i^{\text{shared}}$ of problem instance $\boldsymbol{x}_i$.

## 5 Experiments

We assess the effectiveness of PARCO on representative multi-agent CO problems, spanning both routing and scheduling domains. Specifically, we evaluate its performance on two challenging routing problems – the min-max *heterogeneous capacitated vehicle routing problem* (HCVRP) and the *open multi-depot capacitated pickup and delivery problem* (OMDCPDP) – as well as a scheduling problem, the *flexible flow shop problem* (FFSP). Experimental details are available through Appendix B[3].

### 5.1 Experimental Settings

**HCVRP** *Problem.* The min-max HCVRP involves $M$ agents serving customer demands while adhering to heterogeneous vehicle capacity constraints. Each vehicle can replenish its load by returning to the depot. The objective is to minimize the longest route taken by any agent (min-max), ensuring balanced workload distribution. *Traditional solvers.* We include state-of-the-art SISRs [12], Genetic Algorithm (GA) [39] and Simulated Annealing (SA) [32]. *Neural baselines.* We evaluate the sequential planning baselines Attention Model (AM) [47], Equity Transformer (ET) [84] and Decoupling Partition and Navigation (DPN) [104], and autoregressive models with agent selection $\text{DRL}_{Li}$ [55] and state-of-the-art learning method 2D-Ptr [63]. Additional problem and experimental details are available in Appendix A.1 and Appendix B.1, respectively.

**OMDCPDP** *Problem.* The OMDCPDP is a challenging problem arising in last-mile delivery settings where $M$ agents starting from different locations (i.e., multiple depots) must pick up and deliver parcels without returning to their starting point (i.e., open). Agents have a capacity constraint for orders that can be carried out as a stacking limit: a tour can include more pickups than the constraint, but the agent must deliver corresponding orders so that carrying capacity is freed. The goal is to minimize the lateness, i.e., the sum of delivery arrival times. *Traditional solvers.* We include the popular and efficient optimization suite Google OR-Tools [19] as a classical baseline. *Neural baselines.* We evaluate models specializing in pickup and delivery problems, including the autoregressive Heterogeneous Attention Model (HAM) [54] for sequential planning and MAPDP [108] for parallel planning. Additional problem and experimental details are available in Appendix A.2 and Appendix B.2, respectively.

**FFSP** *Problem.* In FFSP, $N$ jobs must be processed by $M$ machines divided equally in $S$ stages. Jobs follow a specified sequence through these stages. Within each, any available machine can process the job, with the key constraint that no machine can handle multiple jobs simultaneously. The goal is to schedule the jobs so that all jobs are finished in the shortest time possible. *Traditional solvers.* We incorporate the widely used and powerful Gurobi solver [24] as a baseline. Furthermore, we include dispatching rules Random and Shortest Job First (SJF), Particle Swarm Optimization (PSO) [81] and Genetic Algorithm (GA) [25]. *Neural baselines.* Notable benchmarks include the Matrix Encoding Network (MatNet) [50] which demonstrates superior performance on FFSP. Additional problem and experimental details are available in Appendix A.3 and Appendix B.3, respectively.

---

[3]Source code is available at `https://github.com/ai4co/parco`

Table 1: Main results on different problems with different configurations for problem size $N$ and number of agents $M$. For all metrics, the lower the better ($\downarrow$).

**HCVRP**

| | | | | | | | | | | | | | | | | | | |
|---|---|---|---|---|---|---|---|---|---|---|---|---|---|---|---|---|---|---|
| N | 60 | | | | | | | | | 100 | | | | | | | | |
| M | 3 | | | 5 | | | 7 | | | 3 | | | 5 | | | 7 | | |
| Metric | Obj. | Gap | Time | Obj. | Gap | Time | Obj. | Gap | Time | Obj. | Gap | Time | Obj. | Gap | Time | Obj. | Gap | Time |
| SISRs | 6.57 | 0.00% | 271s | 4.00 | 0.00% | 274s | 2.91 | 0.00% | 276s | 10.29 | 0.00% | 615s | 6.17 | 0.00% | 623s | 4.45 | 0.00% | 625s |
| GA | 9.21 | 40.18% | 233s | 6.89 | 72.25% | 320s | 5.98 | 105.50% | 405s | 15.33 | 48.98% | 479s | 10.93 | 77.15% | 623s | 9.10 | 104.49% | 772s |
| SA | 7.04 | 7.15% | 130s | 4.39 | 9.75% | 289s | 3.30 | 13.40% | 362s | 11.13 | 8.16% | 434s | 6.80 | 10.21% | 557s | 5.01 | 12.58% | 678s |
| AM (g.) | 8.49 | 29.22% | 0.08s | 5.51 | 37.75% | 0.08s | 4.15 | 42.61% | 0.09s | 12.68 | 23.23% | 0.14s | 8.10 | 31.28% | 0.13s | 6.13 | 37.75% | 0.13s |
| ET (g.) | 7.58 | 15.37% | 0.15s | 4.76 | 19.00% | 0.17s | 3.58 | 23.02% | 0.16s | 11.74 | 14.09% | 0.25s | 7.25 | 17.50% | 0.25s | 5.23 | 17.53% | 0.26s |
| DPN (g.) | 7.50 | 14.16% | 0.18s | 4.60 | 15.00% | 0.19s | 3.45 | 18.56% | 0.26s | 11.54 | 12.15% | 0.30s | 6.94 | 12.48% | 0.40s | 4.98 | 11.91% | 0.43s |
| DRL$_{Li}$ (g.) | 7.43 | 13.09% | 0.19s | 4.71 | 17.75% | 0.22s | 3.60 | 23.71% | 0.25s | 11.44 | 11.18% | 0.32s | 7.06 | 14.42% | 0.37s | 5.38 | 20.90% | 0.43s |
| 2D-Ptr (g.) | 7.20 | 9.59% | 0.11s | 4.48 | 12.00% | 0.11s | 3.31 | 13.75% | 0.11s | 11.12 | 8.07% | 0.18s | 6.75 | 9.40% | 0.18s | 4.92 | 10.56% | 0.17s |
| PARCO (g.) | **7.12** | 8.37% | 0.04s | **4.40** | 10.00% | 0.05s | **3.25** | 11.68% | 0.05s | **10.98** | 6.71% | 0.06s | **6.61** | 7.13% | 0.05s | **4.79** | 7.64% | 0.05s |
| AM (s.) | 7.62 | 15.98% | 0.14s | 4.82 | 20.50% | 0.13s | 3.63 | 24.74% | 0.14s | 11.82 | 14.87% | 0.29s | 7.45 | 20.75% | 0.28s | 5.58 | 25.39% | 0.28s |
| ET (s.) | 7.14 | 8.68% | 0.21s | 4.46 | 11.50% | 0.22s | 3.33 | 14.43% | 0.22s | 11.20 | 8.84% | 0.41s | 6.85 | 11.02% | 0.38s | 4.98 | 11.91% | 0.40s |
| DPN (s.) | 7.08 | 7.76% | 0.25s | 4.35 | 8.75% | 0.28s | 3.20 | 9.97% | 0.38s | 11.04 | 7.29% | 0.48s | 6.66 | 7.94% | 0.52s | 4.79 | 7.64% | 0.78s |
| DRL$_{Li}$ (s.) | 6.97 | 6.09% | 0.30s | 4.34 | 8.50% | 0.36s | 3.25 | 11.68% | 0.43s | 10.90 | 5.93% | 0.60s | 6.65 | 7.78% | 0.76s | 4.98 | 11.91% | 0.92s |
| 2D-Ptr (s.) | 6.82 | 3.81% | 0.13s | 4.20 | 5.00% | 0.13s | 3.09 | 6.19% | 0.14s | 10.71 | 4.08% | 0.22s | 6.46 | 4.70% | 0.23s | 4.68 | 5.17% | 0.24s |
| PARCO (s.) | **6.82** | 3.81% | 0.05s | **4.17** | 4.25% | 0.05s | **3.06** | 5.15% | 0.07s | **10.61** | 3.11% | 0.08s | **6.36** | 3.08% | 0.08s | **4.58** | 2.92% | 0.09s |

**OMDCPDP**

| | | | | | | | | | | | | | | | | | | |
|---|---|---|---|---|---|---|---|---|---|---|---|---|---|---|---|---|---|---|
| N | 50 | | | | | | | | | 100 | | | | | | | | |
| M | 5 | | | 7 | | | 10 | | | 10 | | | 15 | | | 20 | | |
| Metric | Obj. | Gap | Time | Obj. | Gap | Time | Obj. | Gap | Time | Obj. | Gap | Time | Obj. | Gap | Time | Obj. | Gap | Time |
| OR-Tools | 37.61 | 4.61% | 30s | 30.18 | 3.38% | 30s | 24.48 | 1.79% | 30s | 66.78 | 4.99% | 60s | 51.49 | 3.15% | 60s | 43.90 | 1.53% | 60s |
| HAM (g.) | 39.67 | 10.30% | 0.11s | 31.49 | 7.85% | 0.11s | 29.24 | 21.23% | 0.13s | 71.12 | 11.60% | 0.24s | 54.31 | 8.69% | 0.27s | 53.73 | 23.93% | 0.29s |
| MAPDP (g.) | 37.36 | 4.09% | 0.02s | 30.36 | 4.08% | 0.01s | 24.88 | 3.46% | 0.01s | 66.54 | 4.71% | 0.01s | 52.08 | 4.34% | 0.01s | 44.71 | 3.40% | 0.01s |
| PARCO (g.) | **37.27** | 3.84% | 0.02s | **30.12** | 3.27% | 0.01s | **24.72** | 2.81% | 0.01s | **65.85** | 3.67% | 0.02s | **51.45** | 3.11% | 0.01s | **44.46** | 2.84% | 0.01s |
| HAM (s.) | 36.26 | 1.17% | 1.18s | 29.54 | 1.38% | 1.31s | 27.77 | 15.24% | 1.37s | 66.91 | 5.29% | 2.33s | 51.60 | 3.42% | 2.53s | 52.24 | 20.52% | 2.72s |
| MAPDP (s.) | 35.64 | 0.02% | 0.02s | 29.03 | 0.23% | 0.02s | 23.97 | 0.35% | 0.01s | 63.64 | 0.65% | 0.03s | 50.07 | 0.75% | 0.03s | 43.57 | 1.13% | 0.02s |
| PARCO (s.) | **35.64** | 0.00% | 0.03s | **28.96** | 0.00% | 0.02s | **23.89** | 0.00% | 0.02s | **63.20** | 0.00% | 0.04s | **49.69** | 0.00% | 0.03s | **43.08** | 0.00% | 0.03s |

**FFSP**

| | | | | | | | | | | | | | | | | | | |
|---|---|---|---|---|---|---|---|---|---|---|---|---|---|---|---|---|---|---|
| N | 20 | | | 50 | | | 100 | | | 50 | | | | | | | | |
| M | 12 | | | | | | | | | 18 | | | 24 | | | 30 | | |
| Metric | Obj. | Gap | Time | Obj. | Gap | Time | Obj. | Gap | Time | Obj. | Gap | Time | Obj. | Gap | Time | Obj. | Gap | Time |
| Gurobi (1m) | 35.29 | 42.4% | 60s | - | - | 60s | - | - | 60s | - | - | 60s | - | - | 60s | - | - | 60s |
| Gurobi (10m) | 31.61 | 27.6% | 600s | - | - | 600s | - | - | 600s | - | - | 600s | - | - | 600s | - | - | 600s |
| Random | 47.89 | 93.3% | 0.18s | 93.34 | 89.4% | 0.37s | 167.22 | 86.9% | 0.72s | 67.03 | 112.1% | 0.33s | 54.48 | 130.9% | 0.33s | 46.84 | 137.0% | 0.37s |
| SJF | 31.27 | 26.2% | 0.13s | 56.94 | 15.6% | 0.34s | 99.27 | 11.0% | 0.62s | 38.01 | 20.3% | 0.25s | 29.39 | 24.6% | 0.25s | 24.62 | 24.6% | 0.29s |
| GA | 31.15 | 25.7% | 21s | 56.92 | 15.5% | 44s | 99.25 | 10.9% | 89s | 38.26 | 21.1% | 47s | 29.05 | 16.7% | 50s | 24.52 | 24.1% | 55s |
| PSO | 29.10 | 17.4% | 46s | 55.10 | 11.8% | 82s | 97.3 | 8.8% | 154s | 36.83 | 16.6% | 85s | 28.06 | 12.7% | 89s | 23.44 | 18.6% | 95s |
| MatNet (g.) | 27.26 | 10.0% | 1.22s | 51.52 | 4.6% | 2.17s | 91.58 | 2.4% | 4.97s | 34.82 | 10.2% | 2.42s | 27.52 | 16.7% | 2.65s | 23.65 | 19.7% | 3.09s |
| PARCO (g.) | **26.31** | 6.2% | 0.26s | **51.19** | 3.9% | 0.52s | **91.29** | 2.0% | 0.89s | **32.88** | 4.1% | 0.50s | **24.89** | 5.5% | 0.44s | **20.29** | 2.7% | 0.41s |
| MatNet (s.) | 25.44 | 2.7% | 3.88s | 49.68 | 0.8% | 8.91s | 89.72 | 0.3% | 18s | 33.45 | 5.9% | 9.23s | 26.00 | 10.2% | 9.81s | 22.51 | 13.9% | 11s |
| PARCO (s.) | **24.78** | 0.0% | 0.99s | **49.27** | 0.0% | 1.97s | **89.46** | 0.0% | 4.04s | **31.60** | 0.0% | 1.89s | **23.59** | 0.0% | 1.68s | **19.76** | 0.0% | 1.54s |

## 5.2 Experimental Results

We report the main empirical results for HCVRP, OMDCPDP, and FFSP in Table 1, with average objective function values (Obj.), gaps to the best-known solutions, and inference times for solving each single problem instance. For neural baselines, we evaluate both greedy *(g.)* and sampling *(s.)* performance, using 1280 sampled solutions for routing problems and 128 for FFSP.

In HCVRP, PARCO outperforms all neural baselines in solution quality and speed while providing solutions at a fraction of the solving time required by traditional solvers. In OMDCPDP, our model surpasses all baselines, including OR-Tools. Notably, while the AR baseline HAM struggles with a larger number of agents $M$, PARCO's parallel AR method maintains strong performance across all scales. In FFSP, PARCO outperforms traditional solvers (e.g., Gurobi cannot find solutions in time for $N > 20$), dispatching rules, and MatNet in all tested scenarios while being more than $4\times$ faster. Furthermore, similar to our results in routing, PARCO's advantage becomes even more pronounced in instances with a larger number of agents where PARCO generates higher-quality schedules through effective agent coordination at a fraction of the cost of MatNet.

## 5.3 Analysis

**Effect of Communication Layers**  We showcase the importance of Communication Layers in Fig. 3a. We benchmark different ways to obtain decoder queries (see § 4.5) with 1) No communication (W/o Comm., i.e., with only context features), 2) MLP, 3) MHA, 4) our transformer-based Com-

Table 2: Generalization for unseen numbers of nodes $N$ and agents $M$ (up to $10\times$ those seen during training).

| $N$ | 500 | | | | | | | | | 1000 | | | | | | | | |
|---|---|---|---|---|---|---|---|---|---|---|---|---|---|---|---|---|---|---|
| $M$ | 50 | | | 75 | | | 100 | | | 100 | | | 150 | | | 200 | | |
| Metric | Obj. | Gap | Time | Obj. | Gap | Time | Obj. | Gap | Time | Obj. | Gap | Time | Obj. | Gap | Time | Obj. | Gap | Time |
| OR-Tools | 290.79 | 7.81% | 300s | 223.72 | 4.68% | 300s | 192.59 | 1.51% | 300s | 780.89 | 33.93% | 600s | 642.54 | 36.47% | 600s | 584.27 | 36.81% | 600s |
| HAM ($g.$) | 410.95 | 55.44% | 1.03s | 310.17 | 48.72% | 1.13s | 204.00 | 9.91% | 1.23s | 710.64 | 40.92% | 2.43s | 554.15 | 39.41% | 2.66s | 388.56 | 8.88% | 2.91s |
| PARCO ($g.$) | **268.56** | 1.41% | 0.02s | **211.21** | 1.13% | 0.02s | **187.37** | 0.83% | 0.01s | **510.61** | 0.81% | 0.02s | **401.46** | 0.64% | 0.02s | **359.98** | 0.49% | 0.02s |
| HAM ($s.$) | 409.67 | 54.95% | 1.19s | 305.16 | 46.32% | 1.31s | 203.50 | 9.64% | 1.39s | 708.55 | 40.50% | 2.94s | 552.76 | 39.06% | 3.22s | 384.10 | 7.63% | 3.46s |
| PARCO ($s.$) | **264.38** | 0.00% | 0.03s | **208.56** | 0.00% | 0.02s | **185.61** | 0.00% | 0.02s | **504.30** | 0.00% | 0.79s | **397.51** | 0.00% | 0.91s | **356.87** | 0.00% | 1.26s |

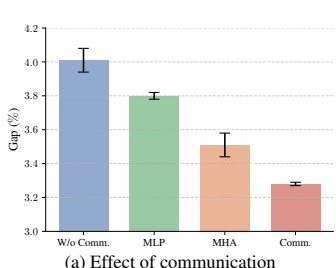

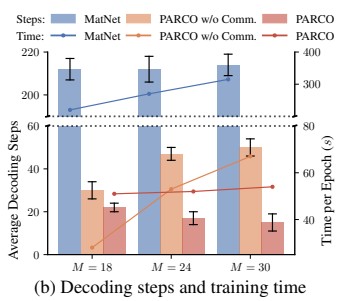

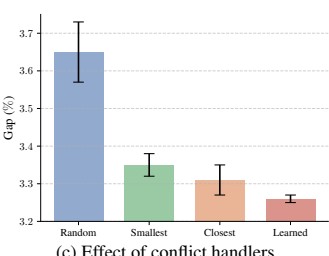

(a) Effect of communication    (b) Decoding steps and training time    (c) Effect of conflict handlers

Figure 3: Analysis of PARCO components.

munication Layers (Comm.). Our Communication Layers consistently outperform other methods. Fig. 3b shows decoding steps and training times on the FFSP for $N = 50$. PARCO greatly reduces the number of steps and training times, with Communication Layers further reducing them through better coordination, especially at a higher number of agents $M$.

**Effect of Conflict Handlers** In Fig. 3c, we compare 1) the random handler from MAPDP and our proposed Priority-based Conflict Handlers in different configurations, namely with priorities based on simple heuristics as 2) "smallest" prioritizing the agent with the lowest cost so far, and 3) "closest", prioritizing agents closer to the corresponding node, and finally 4) "learned" based on model output probabilities. The latter consistently outperforms other methods, which also enjoy a relative reduction in the number of steps for constructing a solution, e.g., with a $4\%$ reduction in conflict rates.

**Large-Scale Generalization** We study the zero-shot large-scale generalization performance of PARCO in the OMDCPDP and report the results in Table 2 for out-of-distribution numbers of nodes $N$ and agents $M$, both up to $10\times$ those seen in training. We find that the AR HAM baseline cannot generalize well to such scales due to the lack of communication and robust parallel construction, while MAPDP cannot be applied to an unseen $M$ because of its inflexible decoder structure.

Conversely, our method outperforms all baselines, including OR-Tools with a 10-minute solving time per instance for $N = 1000$, making PARCO a strong candidate for real-time deployment.

**PARCO vs AR Models Scalability** Finally, we showcase PARCO's speedups against autoregressive methods in Fig. 4. Notably, compared to AR models (e.g., DPN, HAM), PARCO achieves significant speedups of $3.3\times$ up to $24.7\times$, with inference time decreasing as the number of agents $M$ increases: thanks to parallel decoding, fewer solution construction steps for larger $M$ lead to substantially lower latency.

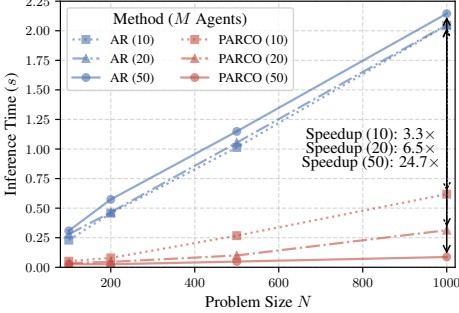

Figure 4: PARCO vs AR inference time. PARCO constructs solutions faster with more agents $M$.

## 6 Conclusion

We introduced PARCO, a learning model to tackle multi-agent combinatorial optimization problems efficiently via parallel autoregressive solution construction. By integrating transformer-based Communication Layers, a Multiple Pointer Mechanism, and Priority-based Conflict Resolution, PARCO enables effective agent coordination and significantly reduces computational latency. Our extensive experiments on multi-agent vehicle routing and scheduling demonstrate that PARCO consistently outperforms state-of-the-art learning-based solvers with better solution quality and higher efficiency.

**Limitations & Future Work**  Although PARCO can efficiently solve multi-agent CO problems with a defined number of agents $M$, it cannot be directly applied to CO tasks where solutions are constructed via an unspecified $M$. In future work, we plan to explore multi-agent CO problems with an unspecified number of agents, which could be achieved by either rolling out a batch of several values of $M$ until an optimal solution is reached or by employing a prediction module to predict an optimal agent number. We defer additional discussions to Appendix C.1.

**Acknowledgements**  We are deeply grateful to the members of the AI4CO open research community for their invaluable contributions to PARCO and related projects, including RL4CO. We also thank the anonymous reviewers who greatly helped improve our paper with their constructive feedback. This work was supported by the Institute of Information & Communications Technology Planning & Evaluation (IITP) grant, funded by the Korean government (MSIT) [Grant No. 2022-0-01032, Development of Collective Collaboration Intelligence Framework for Internet of Autonomous Things]; National Research Foundation of Korea(NRF) grants funded by the Korea government(MSIT) (No. RS-2024-00410082 and No. RS-2025-00563763), and by the InnoCORE program of the Ministry of Science and ICT(N10250154).

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

# PARCO: Parallel AutoRegressive Models for Multi-Agent Combinatorial Optimization
## *Supplementary Material*

## A  Problem Definitions

### A.1  HCVRP

The min-max HCVRP (Heterogeneous Capacitated Vehicle Routing Problem) consists of $M$ agents sequentially visiting customers to satisfy their demands, with constraints including each customer can be visited exactly once and the amount of demand satisfied by a single vehicle in a trip cannot exceed its capacity, which can be reloaded by going back to the depot. The goal is to minimize the makespan, i.e., the worst route.

Consider a problem with $N + 1$ nodes (including $N$ customers and a depot) and $M$ vehicles. The depot is indexed as 0, and customers are indexed from 1 to $N$.

**Indices**
$i, j$        Node indices, where $i, j = 0, \ldots, N$ (0 represents the depot)
$k$        Vehicle index, where $k = 1, \ldots, M$

**Parameters**
$N$        Number of customer nodes (excluding depot)
$M$        Number of vehicles
$X_i$        Location of node $i$
$d_i$        Demand of node $i$ ($d_0 = 0$ for the depot)
$Q_k$        Capacity of vehicle $k$
$f_k$        Speed of vehicle $k$
$c_{ij}$        Distance between nodes $i$ and $j$

**Decision Variables**
$x_{ijk}$        $\begin{cases} 1 & \text{if vehicle } k \text{ travels directly from node } i \text{ to node } j \\ 0 & \text{otherwise} \end{cases}$
$l_{ijk}$        Remaining load of vehicle $k$ before travelling from node $i$ to node $j$

**Objective Function:**

$$\min_{} \max_{k=1,\ldots,m} \left( \sum_{i=0}^{N} \sum_{j=0}^{N} \frac{c_{ij}}{f_k} x_{ijk} \right) \tag{9}$$

**Subject to:**

$$\sum_{k=1}^{m}\sum_{j=0}^{N} x_{ijk} = 1 \qquad\qquad i = 1, \ldots, N \tag{10}$$

$$\sum_{i=0}^{N} x_{ijk} - \sum_{h=0}^{N} x_{jhk} = 0 \qquad\qquad j = 0, \ldots, N, \ k = 1, \ldots, m \tag{11}$$

$$\sum_{k=1}^{m}\sum_{i=0}^{N} l_{ijk} - \sum_{k=1}^{m}\sum_{h=0}^{N} l_{jhk} = d_j \qquad\qquad j = 1, \ldots, N \tag{12}$$

$$d_j x_{ijk} \leq l_{ijk} \leq (Q_k - d_i) \cdot x_{ijk} \qquad i, j = 0, \ldots, N, \ k = 1, \ldots, m \tag{13}$$

$$x_{ijk} \in \{0, 1\} \qquad i, j = 0, \ldots, N, \ k = 1, \ldots, m \tag{14}$$

$$l_{ijk} \geq 0, d_i \geq 0 \qquad i, j = 0, \ldots, N, \ k = 1, \ldots, m \tag{15}$$

**Constraint Explanations:** The formulation is subject to several constraints that define the feasible solution space. Eq. (10) ensures that each customer is visited exactly once by one vehicle. The flow conservation constraint (11) guarantees that each vehicle that enters a node also leaves that node, maintaining route continuity. Demand satisfaction is enforced by constraint (12), which ensures that the difference in load before and after serving a customer equals the customer's demand. The vehicle capacity constraint (13) ensures that the load carried by a vehicle does not exceed its capacity and is sufficient to meet the next customer's demand.

## A.2 OMDCPDP

The OMDCPDP (Open Multi-Depot Capacitated Pickup and Delivery Problem) is a practical variant of the pickup and delivery problem in which agents have a stacking limit of orders that can be carried at any given time. Pickup and delivery locations are paired, and pickups must be visited before deliveries. Multiple agents start from different depots without returning (open). The goal is to minimize the sum of arrival times to delivery locations, i.e., minimizing the cumulative lateness.

**Indices**

$i, j$  Node indices, where $i, j = 1, \ldots, 2N$
$k$    Vehicle index, where $k = 1, \ldots, M$

**Sets**

$P$  Set of pickup nodes, $P = \{1, \ldots, N\}$
$D$  Set of delivery nodes, $D = \{N + 1, \ldots, 2N\}$

**Parameters**

$N$    Number of pickup-delivery pairs
$M$    Number of vehicles
$c_{ij}$  Travel time between nodes $i$ and $j$
$Q_k$   Capacity (stacking limit) of vehicle $k$
$o_k$   Initial location (depot) of vehicle $k$

**Decision Variables**

$x_{ijk}$  $\begin{cases} 1 & \text{if vehicle } k \text{ travels directly from node } i \text{ to node } j \\ 0 & \text{otherwise} \end{cases}$

$y_{ik}$  $\begin{cases} 1 & \text{if vehicle } k \text{ visits node } i \\ 0 & \text{otherwise} \end{cases}$

$t_i$  Arrival time at node $i$
$l_{ik}$  Load of vehicle $k$ after visiting node $i$

**Objective Function:**

$$\min \sum_{i=N+1}^{2N} t_i \tag{16}$$

**Subject to:**

$$\sum_{k=1}^{m} y_{ik} = 1 \qquad\qquad i = 1, \ldots, 2N \qquad\qquad (17)$$

$$\sum_{j=1}^{2N} x_{o_k,j,k} = 1 \qquad\qquad k = 1, \ldots, m \qquad\qquad (18)$$

$$\sum_{i=1}^{2N} x_{ijk} - \sum_{h=1}^{2N} x_{jhk} = 0 \qquad\qquad j = 1, \ldots, 2N, \ k = 1, \ldots, m \qquad (19)$$

$$y_{ik} = \sum_{j=1}^{2N} x_{ijk} \qquad\qquad i = 1, \ldots, 2N, \ k = 1, \ldots, m \qquad (20)$$

$$t_i + c_{ij} - M(1 - x_{ijk}) \le t_j \qquad\qquad i, j = 1, \ldots, 2N, \ k = 1, \ldots, m \qquad (21)$$

$$t_i \le t_{i+N} \qquad\qquad i \in P \qquad\qquad (22)$$

$$l_{ik} + 1 - M(1 - x_{ijk}) \le l_{jk} \qquad\qquad i \in P, j \ne i + N, k = 1, \ldots, m \qquad (23)$$

$$l_{ik} - 1 + M(1 - x_{ijk}) \ge l_{jk} \qquad\qquad i \in D, j \ne i - N, k = 1, \ldots, m \qquad (24)$$

$$0 \le l_{ik} \le Q_k \qquad\qquad i = 1, \ldots, 2N, \ k = 1, \ldots, m \qquad (25)$$

$$x_{ijk}, y_{ik} \in \{0, 1\} \qquad\qquad i, j = 1, \ldots, 2N, \ k = 1, \ldots, m \qquad (26)$$

$$t_i \ge 0 \qquad\qquad i = 1, \ldots, 2N \qquad\qquad (27)$$

**Constraints Explanations:** Eq. (17) ensures that each node is visited exactly once. Constraint (18) guarantees that each vehicle starts from its designated depot. The flow conservation constraint (19) ensures route continuity for each vehicle. Eq. (20) defines the relationship between x and y variables. Time consistency is enforced by constraint (21), while (22) ensures that pickups are visited before their corresponding deliveries. Constraints (23) and (24) manage the load changes during pickup and delivery operations. Finally, the vehicle capacity constraint (25) ensures that the load never exceeds the vehicle's stacking limit.

**Visualization**  We provide a visualization of a large-scale instance in Fig. 5.

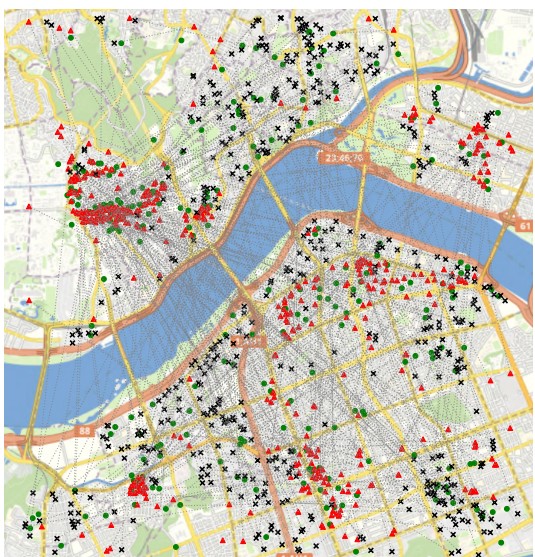

Figure 5: Real-world instance for the OMDCPDP problem in Seoul City, South Korea, with $N = 1000$ locations and $m = 100$ agents (●) showing relations (–) of pickups (▲) and their respective deliveries (✗).

## A.3 FFSP

The flexible flow shop problem (FFSP) is a challenging and extensively studied optimization problem in production scheduling, involving $N$ jobs that must be processed by a total of $M$ machines divided into $i = 1 \ldots S$ stages, each with multiple machines ($m_i > 1$). Jobs follow a specified sequence through these stages, but within each stage, any available machine can process the job, with the key constraint that no machine can handle more than one job simultaneously. The FFSP can naturally be viewed as a multi-agent CO problem by considering each machine as an agent that constructs its own schedule. Adhering to autoregressive CO, agents construct the schedule sequentially, selecting one job (or no job) at a time. The job selected by a machine (agent) at a specific stage in the decoding process is scheduled at the earliest possible time, that is, the maximum of the time the job becomes available in the respective stage (i.e., the time the job finished on prior stages) and the machine becoming idle. The process repeats until all jobs for each stage have been scheduled, and the ultimate goal is to minimize the makespan, i.e., the total time required to complete all jobs.

**Mathematical Model** We use the mathematical model outlined in Kwon et al. [50] to define the FFSP:

**Indices**

| | |
|---|---|
| $i$ | Stage index |
| $j, l$ | Job index |
| $k$ | Machine index in each stage |

**Parameters**

| | |
|---|---|
| $N$ | Number of jobs |
| $S$ | Number of stages |
| $m_i$ | Number of machines in stage $i$ |
| $M$ | A very large number |
| $p_{ijk}$ | Processing time of job $j$ in stage $i$ on machine $k$ |

**Decision variables**

| | |
|---|---|
| $C_{ij}$ | Completion time of job $j$ in stage $i$ |
| $X_{ijk}$ | $\begin{cases} 1 & \text{if job } j \text{ is assigned to machine } k \text{ in stage } i \\ 0 & \text{otherwise} \end{cases}$ |
| $Y_{ilj}$ | $\begin{cases} 1 & \text{if job } l \text{ is processed earlier than job } j \text{ in stage } i \\ 0 & \text{otherwise} \end{cases}$ |

**Objective:**

$$\min \left( \max_{j=1..n} \{ C_{Sj} \} \right) \tag{28}$$

**Subject to:**

$$\sum_{k=1}^{m_i} X_{ijk} = 1 \qquad\qquad i = 1, \ldots, S; \ j = 1, \ldots, N \tag{29}$$

$$Y_{iij} = 0 \qquad\qquad i = 1, \ldots, S; \ j = 1, \ldots, N \tag{30}$$

$$\sum_{j=1}^{N} \sum_{l=1}^{N} Y_{ilj} = \sum_{k=1}^{m_i} \max \left( \sum_{j=1}^{n} (X_{ijk}) - 1, 0 \right) \qquad i = 1, \ldots, S \tag{31}$$

$$Y_{ilj} \leq \max \left( \max_{k=1\ldots m_i} \{X_{ijk} + X_{ilk}\} - 1, 0 \right) \quad i = 1, \ldots, S; \; j, l = 1, 2, \ldots, N \tag{32}$$

$$\sum_{l=1}^{N} Y_{ilj} \leq 1 \quad i = 1, 2, \ldots, S; \; j = 1, 2, \ldots, N \tag{33}$$

$$\sum_{j=1}^{N} Y_{ilj} \leq 1 \quad i = 1, 2, \ldots, S; \; l = 1, 2, \ldots, N \tag{34}$$

$$C_{1j} \geq \sum_{k=1}^{m_1} p_{1jk} \cdot X_{1jk} \quad j = 1, 2, \ldots, N \tag{35}$$

$$C_{ij} \geq C_{i-1j} + \sum_{k=1}^{m_i} p_{ijk} \cdot X_{ijk} \quad i = 2, 3, \ldots, S; \; j = 1, 2, \ldots, N \tag{36}$$

$$C_{ij} + M(1 - Y_{ilj}) \geq C_{il} + \sum_{k=1}^{m_i} p_{ijk} \cdot X_{ijk} \quad i = 1, 2, \ldots, S; \; j, l = 1, 2, \ldots, N \tag{37}$$

**Constraint Explanations:** Here, the objective function Eq. (28) minimizes the makespan of the resulting schedule, that is, the completion time of the job that finishes last. The schedule has to adhere to several constraints: First, constraint set (29) ensures that each job is assigned to exactly one machine at each stage. Constraint sets (30) through (34) define the precedence relationships between jobs within a stage. Specifically, constraint set (30) ensures that a job has no precedence relationship with itself. Constraint set (31) ensures that the total number of precedence relationships in a stage equals $N - m_i$ minus the number of machines with no jobs assigned. Constraint set (32) dictates that precedence relationships can only exist among jobs assigned to the same machine. Additionally, constraint sets (33) and (34) restrict a job to having at most one preceding job and one following job.

Moving on, constraint set (35) specifies that the completion time of a job in the first stage must be at least as long as its processing time in that stage. The relationship between the completion times of a job in consecutive stages is described by constraint set (36). Finally, constraint set (37) ensures that no more than one job can be processed on the same machine simultaneously.

# B   Experimental Details

## B.1   HCVRP

### B.1.1   Baselines

We follow the experimental setup of Liu et al. [63] for baselines, with additional baselines hyperparameter details reported in their respective papers.

**SISR**   The Slack Induction by String Removals (SISR) approach [12] offers a heuristic method for addressing vehicle routing problems (VRPs), focusing on simplifying the optimization process. It combines techniques for route dismantling and reconstruction, along with vehicle fleet minimization strategies. SISR is applied across various VRP scenarios, including those with specific pickup and delivery tasks. In our experiments, we adhere to the hyperparameters provided in the original paper with $\bar{c} = 10, L^{\max} = 10, \alpha = 10^{-3}, \beta = 10^{-2}, T_0 = 100, T_f = 1, \text{iter} = 3 \times 10^5 \times N$.

**GA**  The Genetic Algorithm (GA) [39] is used to address vehicle routing problems (VRPs) and other NP-hard challenges by simulating natural evolutionary processes. GA generates adequate solutions with reasonable computational resources. Our experiment follows the same carefully tuned hyperparameters from [63] with $n = 200, \text{iter} = 40 \times N, P_m = 0.8, P_c = 1$.

**SA**  The Simulated Annealing (SA) method [32] targets the capacitated vehicle routing problem (CVRP) using a population-based approach combined with crossover operators. It incorporates local search and the improved 2-opt algorithm to refine routes alongside crossover techniques to speed up convergence. In our experiment, we follow the same carefully tuned hyperparameters from [63] with $T_0 = 100, T_f = 10^{-7}, L = 20 \times N, \alpha = 0.98$.

**AM**  The Attention Model (AM) [47] applies the attention mechanism to tackle combinatorial optimization problems like the Traveling Salesman and Vehicle Routing Problems. It utilizes attention layers for model improvement and trains using REINFORCE with deterministic rollouts. In our studies, we adopt adjustments from the $\text{DRL}_{Li}$ framework, which involves selecting vehicles sequentially and then choosing the next node for each. Additionally, vehicle-specific features are incorporated into the context vector generation to distinguish between different vehicles.

**ET**  The Equity-Transformer (ET) approach [84] addresses large-scale min-max routing problems by employing a sequential planning approach with sequence generators like the Transformer. It focuses on equitable workload distribution among multiple agents, applying this strategy to challenges like the min-max multi-agent traveling salesman and pickup and delivery problems. In our experiments, we modify the decoder mask in ET to generate feasible solutions for HCVRP and integrate vehicle features into both the input layer and the context encoder, similarly to the setting of Liu et al. [63].

**DPN**  The Decoupling-Partition-Navigation (DPN) approach [104] is a SOTA sequential planning AR baseline that tackles min-max vehicle routing problems (min-max VRPs) by explicitly separating the tasks of customer partitioning and route navigation. It introduces a Partition-and-Navigation (P&N) Encoder to learn distinct embeddings for these tasks, an Agent-Permutation-Symmetric (APS) loss to leverage routing symmetries, and a Rotation-Based Positional Encoding to enhance generalization across different depot locations. We employ a similar setting as ET.

**DRL$_{Li}$**  The DRL approach for solving HCVRP by Li et al. [55] employs a transformer architecture similar to Kool et al. [47] in which the vehicle and node selection happens in two steps via a two selection decoder, thus requiring two actions. We employ their original model with additional context of variable vehicle speeds, noting that in the original setting each model was trained on a single distribution of number of agents $M$, each with always the same characteristics.

**2D-Ptr**  The 2D Array Pointer network (2D-Ptr) [63] addresses the heterogeneous capacitated vehicle routing problem (HCVRP) by using a dual-encoder setup to map vehicles and customer nodes effectively. This approach facilitates dynamic, real-time decision-making for route optimization. Its decoder employs a 2D array pointer for action selection, prioritizing actions over vehicles. The model is designed to adapt to vehicle and customer numbers changes, ensuring robust performance across different scenarios.

### B.1.2  Datasets

**Train data generation**  Neural baselines were trained with the specific number of nodes $N$ and number of agents $M$ they were tested on. In PARCO, we select a varying size and number of customer training schemes: at each training step, we sample $N \sim \mathcal{U}(60, 100)$ and $m \sim \mathcal{U}(3, 7)$. As we show in Table 1, a single PARCO model can outperform baseline models even when they were fitted on a specific distribution. The coordinates of each customer location $(x_i, y_i)$, where $i = 1, \ldots, N$, are sampled from a uniform distribution $\mathcal{U}(0.0, 1.0)$ within a two-dimensional space. The depot location is similarly sampled using the same uniform distribution. The demand $d_i$ for each customer $i$ is also drawn from a uniform distribution $\mathcal{U}(1, 10)$, with the depot having no demand, i.e., $d_0 = 0$. Each vehicle $m$, where $m = 1, \ldots, M$, is assigned a capacity $Q_m$ sampled from a uniform distribution $\mathcal{U}(20, 41)$. The speed $f_m$ of each vehicle is uniformly distributed within the range $\mathcal{U}(0.5, 1.0)$.

**Testing**   Testing is performed on the 1280 instances per $(N, M)$ test setting from Liu et al. [63]. In Table 1, *(g.)* refers to the greedy performance of the model, i.e., taking a single trajectory by taking the maximum action probability; *(s.)* refers to sampling 1280 solutions in the latent space and selecting the one with the lowest cost (i.e., highest reward).

### B.1.3   PARCO Network Hyperparameters

**Encoder**   *Initial Embedding*. This layer projects initial raw features to hidden space. For the depot, the initial embedding is the positional encoding of the depot's location $X_0$. For agents, the initial embedding is the encoding for the initial location, capacity, and speed. *Main Encoder*. we employ $L = 3$ attention layers in the encoder, with hidden dimension $d_h = 128$, 8 attention heads in the MHA, MLP hidden dimension set to 512, with RMSNorm [100] as normalization before the MHA and the MLP.

**Decoder**   *Context Embedding*. This layer projects dynamic raw features to hidden space. The context is the embedding for the depot states, current node states, current time, remaining capacities, time of backing to the depot, and number of visited nodes. *Multiple Pointer Mechanism*. Similarly to the encoder, we employ the same hidden dimension and number of attention heads for the Multiple Pointer Mechanism.

**Communication Layer**   We employ a single transformer layer with hidden dimension $d_h = 128$, 8 attention heads in the MHA, MLP hidden dimension set to 512, with RMSNorm [100] as normalization before the MHA and the MLP. Unlike the encoder layer, which acts between all $M + N$ problem tokens, communication layers are lighter because they communicate between $M$ agents.

**Agent Handler**   We use the Priority-based Conflict Handler guided by the model output probability for managing conflicts: priority is given to the agent whose probability of selecting the conflicting action is the highest (see § 4.6).

### B.1.4   PARCO Training Hyperparameters

Unlike baselines, which are trained and tested on the same distribution, we train a single PARCO model that can effectively generalize over multiple size and agent distributions thanks to our flexible structure. We train PARCO with RL via SymNCO [43] with $K = 10$ symmetric augmentations as shared REINFORCE baseline for 100 epochs using the Adam optimizer [46] with a total batch size 512 (using 4 GPUs in Distributed Data Parallel configuration) and an initial learning rate of $10^{-4}$ with a step decay factor of 0.1 after the 80th and 95th epochs. For each epoch, we sample $4 \times 10^5$ randomly generated data. Training takes around 15 hours in our configuration.

## B.2   OMDCPDP

The setting introduced in OMDCPDP is a more general and realistic setting than the one introduced in Zong et al. [108], particularly due to the multiple depots and the global lateness objective function which is harder to optimize than vanilla min-sum.

### B.2.1   Baselines

**OR-Tools**   Google OR-Tools [19] is an open-source software suite designed to address various combinatorial optimization problems. This toolkit offers a comprehensive selection of solvers suitable for linear programming, mixed-integer programming, constraint programming, and routing and scheduling challenges. Specifically for routing problems like the OMDCPDP, OR-Tools can integrate additional constraints to enhance solution accuracy. For our experiments, we maintained consistent parameters across various problem sizes and numbers of agents. We configured the global span cost coefficient to $10,000$, selected `PATH_CHEAPEST_ARC` as the initial solution strategy, followed by `GUIDED_LOCAL_SEARCH` for local optimization. The solving time was set as $\{30, 60, 300, 600\}$ seconds for $N = \{50, 100, 500, 1000\}$, respectively.

**HAM**   The Heterogeneous Attention Model (HAM) [54] utilizes a neural network-integrated with a heterogeneous attention mechanism that distinguishes between the roles of nodes and enforces

precedence constraints, ensuring the correct sequence of pickup and delivery nodes. This approach helps the deep reinforcement learning model to make informed node selections during route planning. We adapt the original model to handle OMDCPDP with multiple agents akin to the sequential planning of Son et al. [84].

**MAPDP** The Multi-Agent Reinforcement Learning-based Framework for Cooperative Pickup and Delivery Problem (MAPDP) [108] introduces a cooperative PDP with multiple vehicle agents. This framework is trained via a centralized (MA)RL architecture to generate cooperative decisions among agents, incorporating a paired context embedding to capture the inter-dependency of heterogeneous nodes. We adapted the MAPDP to fit our OMDCPDP task, utilizing the same encoder as PARCO to ensure a fair comparison. For the decoder and training phases, we kept the same random conflict handler, and we retained the hyperparameters detailed in the original study overall.

### B.2.2 Datasets

**Train data generation** Neural baselines were trained with the specific number of nodes $N$ and number of agents $M$ they were tested on. In PARCO, we select a varying size and number of customer training schemes: at each training step, we sample $N \sim \mathcal{U}(50, 100)$ and $m \sim \mathcal{U}(5, 20)$. As we show in the Table 1, a single PARCO model can outperform baseline models even when they were fitted on a specific distribution. The coordinates of each customer location $(x_i, y_i)$, where $i = 1, \ldots, N$, are sampled from a uniform distribution $\mathcal{U}(0.0, 1.0)$ within a two-dimensional space. Similarly, we sample $M$ initial vehicle locations from the same distribution. We set the demand $d_i$ for each customer to 1 and the capacity of each vehicle to 3. This emulates realistic settings in which a single package per customer will be picked up and delivered.

**Testing** Testing is performed on 1000 new instances for each setting of in-distribution $N$ and $M$ in Table 1 with the distributions from the training settings. For large-scale generalization in Table 2, we generate 100 new instances.

### B.2.3 PARCO Network Hyperparameters

Most hyperparameters are kept similar to Appendix B.1.3.

**Encoder** *Initial Embedding.* This layer projects initial raw features to hidden space. For depots, the initial embeddings encode the location $o_m$ and the respective vehicle's capacity $Q_m$. For pickup nodes, the initial embeddings encode the location and paired delivery nodes' location. For delivery nodes, the initial embeddings encode the location and paired pickup nodes' location. *Main Encoder.* We employ $l = 3$ attention layers in the encoder, with hidden dimension $d_h = 128$, 8 attention heads in the MHA, MLP hidden dimension set to 512, with RMSNorm [100] as normalization before the MHA and the MLP.

**Decoder** *Context Embedding.* This layer projects dynamic raw features to hidden space. The context is the embedding for the depot states $o_m$, current node states, current length, remaining capacity, and number of visited nodes. These features are then employed to update multiple queries $q_m$, $m = 1, \ldots, M$ simultaneously. *Main Decoder.* Similarly to the encoder, we employ the same hidden dimension and number of attention heads for the Multiple Pointer Mechanism.

**Communication Layer** We employ a single transformer layer with hidden dimension $d_h = 128$, 8 attention heads in the MHA, MLP hidden dimension set to 512, with RMSNorm [100] as normalization before the MHA and the MLP. Note that unlike the encoder layer, which acts between all $M + N$ tokens, communication layers are lighter because they communicate between $M$ agents.

**Agent Handler** We employ the Priority-based Conflict Handler guided by the model output probability for managing conflicts with priority given to the agent whose probability of selecting the conflicting action is the highest (see § 4.6).

### B.2.4 PARCO Training Hyperparameters

For each problem size, we train a single PARCO model that can effectively generalize over multiple size and agent distributions. We train PARCO with RL via SymNCO [43] with $K = 8$ symmetric

augmentations as shared REINFORCE baseline for 100 epochs using the Adam optimizer [46] with a total batch size 128 on a single GPU and an initial learning rate of $10^{-4}$ with a step decay factor of 0.1 after the 80th and 95th epochs. For each epoch, we sample $10^5$ randomly generated data. Training takes less than 5 hours in our configuration.

## B.3 FFSP

### B.3.1 Baselines

**Gurobi** We implement the mathematical model described above in the exact solver Gurobi [24] with a time budget of 60 and 600 seconds per instance. However, with both time budgets, Gurobi is only capable of generating solutions to the FFSP20 instances, similar to the findings made by Kwon et al. [50] for the CPLEX solver.

**Random and Shortest Job First (SJF)** The Random and Shortest Job First (SJF) heuristics are simple construction strategies that build valid schedules in an iterative manner. Starting from an empty schedule, the Random construction heuristic iterates through time steps $t = 0, \ldots T$ and stages $i = 1 \ldots S$ and randomly assigns jobs available at the given time to an idle machine of the respective stage until all jobs are scheduled. Likewise, the SJF proceeds by assigning job-machine pairs with the shortest processing time first.[4]

**Genetic Algorithm (GA)** Genetic Algorithms are metaheuristics widely used by the OR community to tackle the FFSP [38]. The GA iteratively improves multiple candidate solutions called chromosomes. Each chromosome consists of $S \times N$ real numbers, where $S$ is the number of stages and $N$ is the number of jobs. For each job at each stage, the integer part of the corresponding number indicates the assigned machine index, while the fractional part determines job priority when multiple jobs are available simultaneously. Child chromosomes are created through crossover, inheriting integer and fractional parts independently from two parents. Mutations, applied with a 30% chance, use one of four randomly selected methods: exchange, inverse, insert, or change. The implementation uses 25 chromosomes. One initial chromosome is set to the Shortest Job First (SJF) heuristic solution and the best-performing chromosome is preserved across iterations. Each instance runs for 1,000 iterations.

**Particle Swarm Optimization (PSO)** Finally, Particle Swarm Optimization iteratively updates multiple candidate solutions called particles, which are updated by the weighted sum of the inertial value, the local best, and the global best at each iteration [81]. In this implementation, particles use the same representation as GA chromosomes. The algorithm employs 25 particles, with an inertial weight of 0.7 and cognitive and social constants set to 1.5. One initial particle represents the SJF heuristic solution. Like GA, PSO runs for 1,000 iterations per instance.

**MatNet** We benchmark PARCO mainly against MatNet [50], a state-of-the-art NCO architecture for the FFSP. MatNet is an encoder-decoder architecture, which is inspired by the attention model [47]. It extends the encoder of the attention model with a dual graph attention layer, a horizontal stack of two transformer blocks, capable of encoding nodes of different types in bipartite graph-like machines and jobs in the FFSP. Kwon et al. [50] train MatNet using POMO [49].

### B.3.2 Datasets

**Train data generation** We follow the instance generation scheme outlined in Kwon et al. [50] sample processing times for job-machine pairs independently from a uniform distribution within the bounds $[2, 10]$. For the first three FFSP instance types shown in Table 1 we also use the same instance sizes as Kwon et al. [50] with $N = 20, 50$ and $100$ jobs and $M = 12$ machines which are spread evenly over $S = 3$ stages. To test for agent sensitivity in the FFSP, we fix the number of jobs to $N = 50$ but alter the number of agents for the last three instance types shown in Table 1. Still, we use $S = 3$ for this experiment, but alter the number of machines per stage to $M_i = 6, 8$ and $10$, yielding a total of 18, 24 and 30 agents, respectively.

---

[4]To obtain results for the heuristics and metaheuristics, we used the implementation of Kwon et al. [50], provided in the official GitHub repository of the paper: `https://github.com/yd-kwon/MatNet`

**Testing** Testing is performed on 100 separate test instances generated randomly according to the above generation scheme.

### B.3.3 PARCO Network Hyperparameters

**Encoder** To solve the FFSP with our PARCO method, we use a similar encoder as Kwon et al. [50]. The MatNet encoder generates embeddings for all machines of all stages and the jobs they need to process, plus an additional dummy job embedding, which can be selected by any machine in each decoding step to skip to the next step. To compare PARCO with MatNet, we use similar hyperparameters for both models. We use $L = 3$ encoder layers, generating embeddings of dimensionality $d_h = 256$, which are split over $h = 16$ attention heads in the MHA layers. Further, we employ Instance Normalization [87] and a feed-forward network with 512 neurons in the transformer blocks of the encoder.

**Decoder** The machines are regarded as the agents in our PARCO framework. As such, their embeddings are used as queries $q$ in the Multiple Pointer Mechanism Eq. (6), while job embeddings are used as the keys and values. In each decoding step, the machine embeddings are fused with a projection of the time the respective machine becomes idle. Similarly, job embeddings are augmented with a linear transformation of the time they become available in the respective stage before entering the attention head in Eq. (5).

**Communication Layer** We employ a single transformer block with hidden dimension $d_h = 256$ and $h = 16$ attention heads in the MHA, an MLP with 512 hidden units and Instance Normalization.

**Agent Handler** We use the High Probability Handler for managing conflicts: priority is given to the agent whose (log-) probability of selecting the conflicting action is the highest. Formally, priorities $\mathbf{p}_m = \log p_\theta(a_m|x)$ for $m = 1, \ldots, M$.

### B.3.4 PARCO Training Hyperparameters

Regarding the training setup, each training instance $i$ is augmented by a factor of 24, and the average makespan over the augmented instances is used as a shared baseline $b_i^{\text{shared}}$ for the REINFORCE gradient estimator of Eq. (8). We use the Adam optimizer [46] with a learning rate of $4 \times 10^{-4}$, which we alter during training using a cosine annealing scheme. We train separate models for the environment configurations used in Table 1. We train models corresponding to environments with 20 jobs for 100, with 50 jobs for 150 and with 100 jobs for 200 epochs. In each epoch, we train the models using 1,000 randomly generated instances split into batches of size 50.[5]

### B.3.5 Diagram for MatNet Decoding vs. PARCO Decoding for the FFSP

The following figures visualize the decoding for the machines of a given stage using MatNet and PARCO. As one can see in Fig. 6a, MatNet requires a decoder forward pass for each machine to schedule a job on each of them. In contrast, as detailed in Fig. 6b, PARCO can schedule jobs on all machines simultaneously through its Multiple Pointer Mechanism and Agent Handler, leading to significant efficiency gains.

### B.4 Hardware and Software

### B.4.1 Hardware

We experiment on a workstation equipped with 2 INTEL(R) XEON(R) GOLD 6338 CPUs and 8 NVIDIA RTX 4090 graphic cards with 24 GB of VRAM each. Training runs of PARCO take less than 24 hours each. During inference, we employ only one CPU and a single GPU.

### B.4.2 Software

We used Python 3.12, PyTorch 2.5 [78] coupled with PyTorch Lightning [17] with most code based on the RL4CO library [5]. The operating system is Ubuntu 24.04 LTS.

---

[5]Note: to avoid OOMs, for FFSP100 instances, batches are further split into mini-batches of size 25 whose gradients are accumulated.

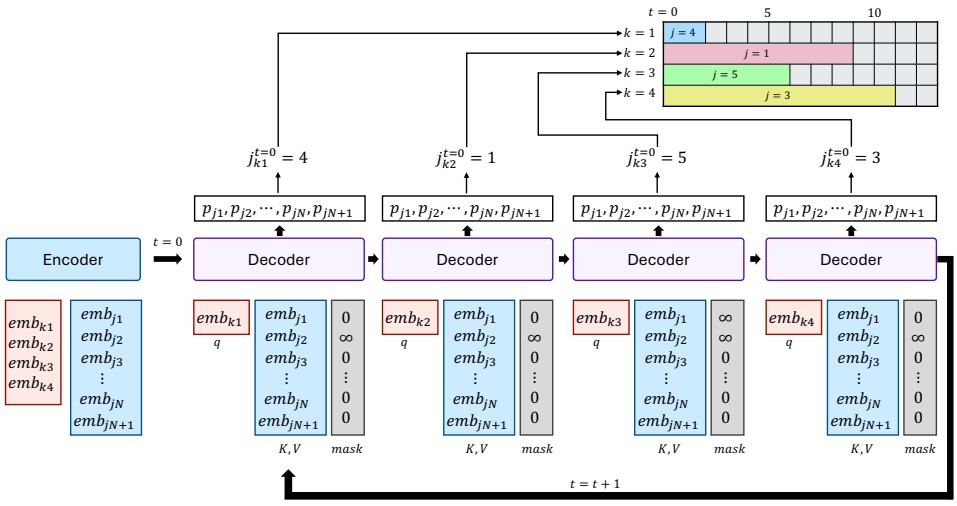

(a) An FFSP Decoding Step with MatNet

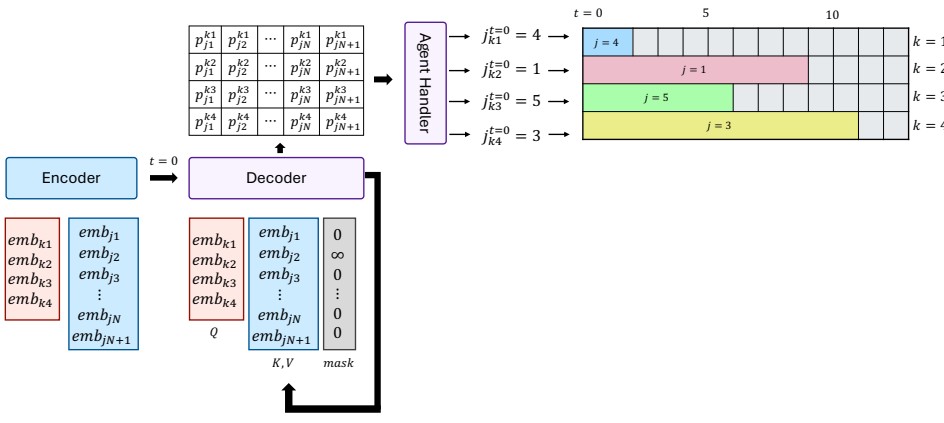

(b) An FFSP Decoding Step with PARCO

Figure 6: Comparison of decoding steps for FFSP with MatNet and PARCO.

## B.5 Source Code

We value open reproducibility and release our source code at 
We also provide the summary of licenses in Table 3.

## C  Additional Materials

### C.1  Further Discussions

**Follow-ups**  We acknowledge there are works based on or inspired by PARCO already out in the wild at the time of publication. These include extending to profiled VRP [29, 30], mixed-shelves picker-routing [67], and self-improvement for better performance [68]. We hope to see even more in the future from diverse research groups, and we remain available for discussions, including through the AI4CO Slack.

**Possible future directions**  In this paragraph, we will share a short list of possible additional future directions for PARCO. One promising direction for NCO is to employ Large Language Models (LLMs) for automating the design of algorithms and particularly heuristics [62, 97, 27]. Several

Table 3: Summary of licenses for used assets.

| Resource | Type | License |
|---|---|---|
| OR-Tools [19] | Code | Apache License, Version 2.0 |
| AM [47] | Code | MIT License |
| ET [84] | Code | Available on Github |
| DPN [104] | Code | MIT License |
| $DRL_{Li}$ [55] | Code | Available on Github |
| 2D-Ptr [63] | Code/Dataset | Available on Github |
| HAM [54] | Code | MIT License |
| Gurobi [24] | Code | Commercial license (free for academic use) |
| MatNet [50] | Code/Dataset | MIT License |
| RL4CO [5] | Library | MIT License |

components of PARCO could be designed by LLMs, including the conflict handling mechanism or attention biases dependent on specific problems [31, 90] as done by Tran et al. [86]. Of interest would also be extensions to other problem variants [99]. Integrating parallel decoding into search methods, as NDS [28], may also yield much more efficient models and enable them to capture different agents.

## C.2 Comparison with Decentralized and Graph-Based Communication Methods

We additionally compare PARCO with decentralized methods and alternative communication models like Graph Neural Networks (GNNs). We experiment on the min-max traveling salesman problem (mTSP), benchmarking PARCO against notable decentralized methods like GNN-DisPN [16], DAN [10] and GNN-based communication models like ScheduleNet [75].

The results, shown in Table 4 with instances from Park et al. [75] as well as in the mTSPLib[6] in Table 5 – additionally adding the HGA solver for further comparison from Mahmoudinazlou and Kwon [72] – demonstrate that PARCO consistently outperforms these methods in solution quality. Centralized training with parallel decoding, as used in PARCO, offers distinct advantages by enabling global coordination and highly efficient solution construction. For example, GNN-based communication in models like ScheduleNet incurs significantly higher computational overhead, making our approach much faster than those methods.

Table 4: Cost comparison on the mTSP with different numbers of salesmen $M$ and number of nodes $N$.

| $N$ | 50 | | | 100 | | | 200 | | | Gap(%) |
|---|---|---|---|---|---|---|---|---|---|---|
| $M$ | 5 | 7 | 10 | 5 | 10 | 15 | 10 | 15 | 20 | |
| LKH3 (solver) | 2.00 | 1.95 | 1.91 | 2.20 | 1.97 | 1.98 | 2.04 | 2.00 | 1.97 | - |
| OR-Tools (solver) | 2.04 | 1.96 | 1.96 | 2.36 | 2.29 | 2.25 | 2.57 | 2.59 | 2.59 | 14.42 |
| GNN-DisPN (g.) | 2.14 | 2.10 | 1.99 | 2.56 | 2.22 | 2.04 | 2.97 | 2.30 | 2.15 | 13.45 |
| DAN (g.) | 2.29 | 2.11 | 2.03 | 2.72 | 2.17 | 2.09 | 2.40 | 2.20 | 2.15 | 11.75 |
| SchedNet (g.) | 2.17 | 2.07 | 1.98 | 2.59 | 2.13 | 2.07 | 2.45 | 2.24 | 2.17 | 10.16 |
| PARCO (g.) | **2.12** | **2.00** | **1.92** | **2.47** | **2.02** | **1.98** | **2.28** | **2.06** | **1.99** | **4.43** |
| DAN (s.) | 2.12 | 1.99 | 1.95 | 2.55 | 2.05 | 2.00 | 2.29 | 2.13 | 2.07 | 6.13 |
| SchedNet (s.) | 2.07 | 1.99 | 1.92 | 2.43 | 2.03 | 1.99 | 2.25 | 2.08 | 2.05 | 4.29 |
| PARCO (s.) | **2.07** | **1.98** | **1.91** | **2.38** | **1.99** | **1.98** | **2.22** | **2.03** | **1.98** | **2.80** |

We note that these experiments refer to a previous version of PARCO in which the conflict handler was not properly working and was similar to the random handler – we would expect retraining PARCO on the newest implementation would perform even better. As the scope of this comparison is to compare against decentralized methods, we do not include more recent approaches as ET [84] or DPN [104] which would outperform this old PARCO version – albeit with slower decoding. We expect that recent follow-ups of our work, which allow for stopping actions by adding special tokens

---

[6]https://profs.info.uaic.ro/mihaela.breaban/mtsplib/MinMaxMTSP/

Table 5: Results for the mTSPLib. CPLEX results with * are optimal solutions. Otherwise, the best-known upper bound of CPLEX results are reported.

| instance_N | eil51 | | | | berlin52 | | | | eil76 | | | | rat99 | | | | Gap (%) |
|---|---|---|---|---|---|---|---|---|---|---|---|---|---|---|---|---|---|
| $M$ | 2 | 3 | 5 | 7 | 2 | 3 | 5 | 7 | 2 | 3 | 5 | 7 | 2 | 3 | 5 | 7 | |
| CPLEX | 222.7* | 159.6 | 124.0 | 112.1 | 4110.2 | 3244.4 | 2441.4 | 2440.9 | 280.9* | 197.3 | 150.3 | 139.6 | 728.8 | 587.2 | 469.3 | 443.9 | 3.40% |
| LKH3 | 222.7 | 159.6 | 124.0 | 112.1 | 4110.2 | 3244.4 | 2441.4 | 2440.9 | 280.9 | 197.3 | 150.3 | 139.6 | 728.8 | 587.2 | 469.3 | 443.9 | 3.40% |
| OR-Tools | 243.3 | 170.5 | 127.5 | 112.1 | 4665.5 | 3311.3 | 2482.6 | 2440.9 | 318.0 | 212.4 | 143.4 | 128.3 | 762.2 | 552.1 | 473.7 | 442.5 | 6.05% |
| HGA | 222.7 | 159.6 | 118.1 | 112.1 | 4110.2 | 3069.6 | 2440.9 | 2440.9 | 280.9 | 196.7 | 142.9 | 127.6 | 666.0 | 517.7 | 450.3 | 436.7 | 0.00% |
| DAN (*s.*) | 252.9 | 178.9 | 128.2 | 114.3 | 5097.7 | 3455.7 | 2677.1 | 2494.5 | 336.7 | 228.1 | 157.9 | 134.5 | 966.5 | 697.7 | 495.6 | 462.0 | 14.51% |
| SchedNet (*s.*) | 239.3 | 173.5 | 125.8 | 112.2 | 4591.6 | 3276.1 | 2517.3 | 2441.4 | 317.7 | 220.8 | 153.8 | 131.7 | 781.2 | 627.1 | 502.3 | 464.4 | 8.55% |
| PARCO (*s.*) | 231.7 | 170.8 | 123.6 | 112.5 | 4429.2 | 3331.6 | 2519.3 | 2444.8 | 295.7 | 202.9 | 147.7 | 128.6 | 762.4 | 581.4 | 473.5 | 450.7 | 5.22% |

to the embedding, as MACSIM [68], would outperform ET and DPN, perhaps considerably. We leave this as an interesting direction for future work.

## C.3 Convergence Rates

Table 6: Convergence rates in different problems: cost as a percentage of training budget.

| | 10% | 25% | 50% | 75% | 100% |
|---|---|---|---|---|---|
| HCVRP | $5.13 \pm 0.09$ | $5.00 \pm 0.06$ | $4.90 \pm 0.05$ | $4.88 \pm 0.03$ | $4.79 \pm 0.03$ |
| OMDCPDP | $46.23 \pm 0.21$ | $45.57 \pm 0.15$ | $45.34 \pm 0.11$ | $44.87 \pm 0.10$ | $44.48 \pm 0.09$ |
| FFSP | $95.45 \pm 0.52$ | $94.32 \pm 0.31$ | $92.88 \pm 0.20$ | $92.15 \pm 0.12$ | $91.48 \pm 0.08$ |

Table 6 shows the convergence rate of PARCO across different problems on validation datasets at different training budgets (in percentage). PARCO is robust during training and converges stably.

## C.4 XXL Instances

To further strengthen our results, we have tested PARCO for even larger scales in both the number of nodes and the number of agents. We generated 16 new instances for $N = 5000$ nodes and 3 different values of $M$ agents (a total of 48 instances) of OMDCPDP. We evaluated OR-Tools with 1 hour of runtime and greedy performance for HAM and PARCO.

Table 7: Large-scale generalization results for OMDCPDP with $N = 5000$.

| | $M = 500$ | | | $M = 750$ | | | $M = 1,000$ | | |
|---|---|---|---|---|---|---|---|---|---|
| | Obj. | Gap | Time | Obj. | Gap | Time | Obj. | Gap | Time |
| OR-Tools | 5575.73 | 134.06% | 3600s | 5127.46 | 115.24% | 3600s | 4974.81 | 188.10% | 3600s |
| HAM | 4813.99 | 102.08% | 17.4s | 3732.06 | 97.33% | 19.5s | 3258.26 | 88.69% | 22.3s |
| **PARCO** | **2382.22** | **0.0%** | **0.21s** | **1891.28** | **0.0%** | **0.21s** | **1726.78** | **0.0%** | **0.22s** |

As shown in Table 7, PARCO excels at generalization at XXL scales with $50\times$ the number of nodes and agents seen during training and up to $1,000$ agents. Thanks to its massively parallel structure, PARCO can solve such instances in a fraction of a second with better results than OR-Tools – at a $10,000\times$ speedup. This makes PARCO ideal for real-world, real-time, large-scale complex problems.

