# OpenReview forum: "PARCO: Parallel AutoRegressive Models for Multi-Agent Combinatorial Optimization"
_NeurIPS.cc/2025/Conference — NeurIPS 2025 poster_

### Official Review · Reviewer_GvYn · 2025-06-28

**Clarity:** 2
**Significance:** 2
**Originality:** 3
**Rating:** 4
**Confidence:** 2

**Summary:**

The paper introduces **PARCO**, a reinforcement learning framework for multi-agent combinatorial optimization that integrates transformer-based communication, parallel decision-making, and conflict resolution. The approach achieves improved coordination, generalization, and computational efficiency on tasks such as vehicle routing and scheduling.

**Questions:**

1. Since the joint action space grows with the number of agents, how does this affect learning stability and computational complexity?
2. Is the agent embedding hah_aha also used in the decoding process (Equations 5, 6, and 7)?
3. The agent priority is derived from the policy π\piπ, which may introduce an overestimation bias. Could this cause high-priority agents to be reinforced more strongly, and how is this mitigated?
4. For the generalization results, how is PARCO used in an untrained number of agents?  Is there a training-testing mismatch?

**Ethical Concerns:**

["NO or VERY MINOR ethics concerns only"]

**Final Justification:**

The paper proposes a multi-agent method to solve a realistic robotic task. The initial submission missed details on training performances, which were added in the rebuttal. I have raised the score to 4.

**Limitations:**

See Question 1.

**Quality:**

3

**Strengths And Weaknesses:**

**Strengths:**

- The reformulation of the multi-agent decision-making problem is theoretically sound and enables more efficient parallel execution.
- The experimental evaluation is thorough, comparing against a broad set of baselines, including both classical optimization methods and neural-based approaches.

**Weaknesses:**

- The proposed formulation allows multiple agents to act simultaneously, whereas the original problem restricts movement to one agent at a time. It is unclear whether efficiency comparisons across these settings are entirely fair, given this difference in assumptions.
- The training curves are missing from the main paper. It would be helpful to include them to evaluate convergence behavior. Additionally, it is unclear whether Figure 3b represents wall-clock training time or another metric — this should be clarified.

---

> ### Author Rebuttal · Authors · 2025-07-31
>
> Thank you for your constructive feedback and for your time reviewing our paper. We greatly appreciate your acknowledgment of PARCO's strengths, including our model's efficacy across multiple settings and the thorough experimental design.
>
> We will address your concerns and questions in the following.
>
> **W1: About fair efficiency comparison**
>
> This is a crucial point that highlights a core contribution of our work. The reformulation from a sequential to a parallel decision-making process is an intentional innovation designed to find higher-quality solutions more efficiently.
>
> The comparison is fair for two key reasons:
>
> - Shared Objective: All models are ultimately judged on the quality of the final, complete solution (e.g., the total makespan or the longest route) for the same underlying optimization problem. The method of constructing that solution is what differs.
>
> - Efficiency as a Contribution: PARCO's ability to construct this solution in parallel is precisely the source of its efficiency advantage. It drastically reduces the number of sequential decoding steps required to build a valid solution (see figure 3b), leading to significant speedups in both training and inference without sacrificing - and often improving - the final solution quality.
>
> Therefore, the efficiency gain isn't an artifact of an unfair comparison; it's the direct result of a more advanced and arguably more realistic solution construction methodology.
>
> **W2: About training curves and clarification of figure 3b**
>
> Thanks for pointing us to missing information and training details. Regarding the convergence analysis, while note that we cannot add figures or any other media due to the rebuttal policy, we will provide the below table with PARCO's convergence evaluated on the validation datasets of the respective problems  (Training budgets refer to the percentage of the total training epochs):
>
> | Training budget  | 10% | 25% | 50% | 75% | 100% |
> |------|-----|-----|-----|-----|------|
> | HCVRP ($N=100$, $M=7$) | 5.13 ± 0.09 | 5.00 ± 0.06 | 4.90 ± 0.05 | 4.88 ± 0.03 | 4.79 ± 0.03 |
> | OMDCPDP ($N=100$, $M=20$) | 46.23 ± 0.21 | 45.57 ± 0.15 | 45.34 ± 0.11 | 44.87 ± 0.10 | 44.48 ± 0.09 |
> | FFSP ($N=100$, $M=12$) | 95.45 ± 0.52 | 94.32 ± 0.31 | 92.88 ± 0.20 | 92.15 ± 0.12 | 91.48 ± 0.08 |
>
> We can observe that PARCO is robust and yields consistent improvements.
>
> To clarify, the "Time per Epoch (s)" metric on the right y-axis of Figure 3b represents the wall-clock training time in seconds. This is a standard measure of computational throughput, indicating how long it takes to complete one full training epoch on the FFSP dataset. The figure illustrates that PARCO's parallel architecture allows it to process an entire epoch of training data significantly faster than the sequential MatNet baseline, demonstrating its superior training efficiency.
>
>
> **Q1: Learning stability and computational complexity**
>
> The learning stability and computational complexity with growing number of agents is one of the greatest strengths of PARCO. PARCO's design directly confronts the challenge of a large joint action space via:
> - Factorization of joint action space: PARCO does not compute a probability distribution over the entire joint action space of size $N^M$. Instead, PARCO leverages the fact that all agents share the same action space, generating a joint logit space of dimension $N \times M$. In addition, our method generates this joint logit space in parallel (one forward pass) via our multiple-pointer mechanism enabling efficient scaling to multi-agent settings without combinatorial explosion.
> - Coordination through communication layers: The attention-based coordination layer effectively helps to coordinate even a large (and as we show in the result table below also an unseen) number of agents
> - Intelligently resolving decision conflicts (e.g., two agents selecting the same customer) using learned priorities
>
>
> The effectiveness of these components is also demonstrated in Figure 3a, 3b, and 3c of the paper. While 3a and 3c show the effectiveness of our communication layer and conflict handler respectively, figure 3b demonstrates that PARCO requires fewer forward passed of the neural policy to construct a solution, when the number of agents increases, leading to much shorter training and inference times.
>
>
> **Q2: Is the agent embedding also used in the decoding process?**
>
> Yes, after the communication layer, which sends the agent embeddings as well as context information (like remaining workload) through a transformer layer, the resulting query vectors q are used in the decoding process in order to generate the score matrix $\mathbf{u} \in \mathbb{R}^{M \times N}$ (i.e. one score for each agent-node pair).
>
>
> **Q3: About the overestimation bias**
>
> Thank you for the insightful question. While priority is derived from the policy, our method does not introduce overestimation bias, due to the structure of the REINFORCE + POMO update and our deterministic fallback mechanism.
>
> #### Gradient-Based Analysis:
>
> In case of conflict (i.e., multiple agents selecting the same node), only the agent with the highest action probability $g_{\theta} (a_t^m | a_{<t}, \mathbf{h})$ is allowed to proceed; the others fall back to a deterministic "do-nothing" action (with $\log p_\theta (a_t^m) = 0$ and zero gradient).
> We apply REINFORCE with a POMO-style baseline. The policy gradient for solution $\tau_i$ is:
>
> $$
> \nabla_\theta J(\theta) \propto \nabla_\theta \log \pi_\theta(\tau_i|x) \cdot (R(\tau_i) - \bar{R})
> $$
>
> where $R$ is the return the $i$-th POMO rollout and $\bar{R}$ is the mean return across multiple POMO rollouts.
>
> ##### Case 1: Agent $m$ wins in a conflict and executes $a_t^m$ in solution $\tau_i$:
>
> - If $R_i > \bar{R}$: the update is positive &#8594; the action is reinforced and the chances of agent $m$ winning increase.
> - If $R_i < \bar{R}$: the update is negative &#8594; the action is penalized and the chances of agent $m$ winning decrease.
> Thus, high-priority agents are not blindly reinforced; they are only updated in proportion to actual performance.
>
> ##### Case 2: The agent looses and is forced into a deterministic fallback action with probability 1:
>
> - $\log \pi(\text{fallback}) = 0$
> - $\nabla_\theta \log \pi = 0$
>
> Hence, the gradient update is zero, and no reinforcement occurs, even if the original action had high $\pi$. Therefore, no overestimation bias is introduced. The priority mechanism acts only as a selection filter and the gradient remains aligned with actual performance outcomes.
>
>
> **Q4: Is PARCO used in an untrained number of agents?**
>
> Yes! We also tested PARCO in not only a new number of agents $M$ , but also new number of nodes $N$. This is the result of Table 2.
>
>
> Moreover, we have further tested PARCO for even larger scales in both number of nodes and number of agents during the rebuttal. We generated 16 new instances for $N=5000$ nodes and 3 different values of $M$ agents total of 48 instances of OMDCPDP and evaluate OR-Tools with 1 hour of runtime and greedy performance for HAM and PARCO. The results are in the table below:
>
>
> |        | $M=500$ |  |  | $M=750$ |  |  | $M=1,000$ |  |  |
> |--------|-------------|-------------|-------------|-------------|-------------|-------------|-------------|-------------|-------------|
> |        | Obj. | Gap | Time | Obj. | Gap | Time | Obj. | Gap | Time |
> | OR-Tools | 5575.73 | 134.06% | 3600s | 5127.46 | 115.24% | 3600s | 4974.81 | 188.10% | 3600s |
> | HAM | 4813.99 | 102.08% | 17.4s | 3732.06 |97.33%| 19.5s | 3258.26 | 88.69% | 22.3s |
> | PARCO | **2382.22** | **0.0%** | **0.21s** | **1891.28** | **0.0%** | **0.21s** | **1726.78** | **0.0%** | **0.22s** |
>
> As we can see from the results, PARCO excels at generalization in extremely large scales with $50\times$ the number of nodes and agents seen during training and up to a whopping $1,000$ agents. Thanks to its massively parallel structure, PARCO can solve such instances in a fraction of a second, with better results than OR-Tools at a $10,000\times$ speedup. This makes our PARCO ideal for real-world, real-time, large-scale complex applications.

---

### Official Review · Reviewer_NRcw · 2025-06-29

**Clarity:** 3
**Significance:** 1
**Originality:** 1
**Rating:** 3
**Confidence:** 4

**Summary:**

The authors address two major limitations of the existing autoregressive paradigm for multi-agent combinatorial optimization problems, namely inefficiency and lack of coordination. By introducing a communication layer to enhance coordination, a multiple-point mechanism to improve inference speed, and a conflict handler to ensure constraint satisfaction, the proposed method achieves certain improvements. However, there remain several aspects that require further clarification, including the motivation, problem definition, effectiveness of the conflict handler, and the presentation of experimental results.

**Questions:**

1. Could the authors provide more precise mathematical definitions for key concepts such as "agent," "node," decision variables, and optimization objectives? How do these definitions clarify the differences and relationships among multi-agent CO, single-agent CO, and conventional CO problems?

2. The conflict handler is implemented as a fixed, non-adaptive fallback mechanism. Have the authors considered or experimented with adaptive or learnable conflict resolution methods? Additionally, could the authors provide more detailed analysis and experimental results on the selection of the fallback action, its optimality, and its role during different training phases?

3. The paper currently does not include training curve plots. Could the authors provide these curves to help assess the convergence behavior, stability, and training efficiency of the proposed method?

4. Given that the performance improvements reported are relatively marginal, could the authors further discuss the practical significance of their approach and clarify whether the addressed motivation reflects a critical challenge in combinatorial optimization?

**Ethical Concerns:**

["NO or VERY MINOR ethics concerns only"]

**Final Justification:**

After the rebuttal, the authors have addressed my concerns regarding the Conflict Handler and some aspects of the problem definition, and have also stated that they will include training curves. However, I still have some reservations about the clarity of the writing.

**Limitations:**

yes

**Quality:**

2

**Strengths And Weaknesses:**

**Strength**

The overall quality of this paper is commendable. It presents a motivation, targeted model design, and extensive experimental comparisons. The writing in most parts of the paper is quite clear and easy-to-understand.

**Weakness**
1. The paper lacks clear and precise definitions regarding the multi-agent CO problem. Specifically, the concepts of "agent," the justification for modeling the problem as multi-agent, as well as definitions for "node," the relationships between nodes, decision variables, and optimization objectives are not rigorously formulated. This makes it difficult for readers to quickly grasp the differences, connections, and levels of complexity among multi-agent CO, single-agent CO, and conventional CO problems.
2. The literature review is not sufficiently comprehensive, which undermines the persuasiveness of the motivation. Additionally, the definition of the CO problem under study may be overly broad, making the proposed motivation less convincing in many scenarios. For example, in applications such as urban ride-hailing and task assignment, drivers, passengers, and workers are heterogeneous; however, many existing works adopt a single-agent RL paradigm and, combined with algorithms such as the KM algorithm, can guarantee conflict-free assignments and make decisions within hundreds of milliseconds even at large urban scale. These findings contradict the premises of the authors’ motivation.
3. The effectiveness of the proposed conflict handler appears to be very limited. In my experience, it is extremely challenging for neural networks to consistently satisfy nested, diverse, large-scale, and overlapping constraints. The handler adopted by the authors relies on a fixed, non-adaptive, and non-learnable fallback mechanism, which makes it difficult to guarantee constraint satisfaction unless every action fall backs to r. Therefore, more explicit experimental results and discussion are needed regarding the choice of r, its optimality, its role during training, the specific phases in which it is necessary or unnecessary, and when constraints can be satisfied without its assistance. Such analysis is essential to justify the necessity and effectiveness of the proposed handler.
4. The experimental results lack training curve plots, making it difficult to assess the convergence, stability, training speed, and computational time of the proposed approach. Additionally, the reported performance improvements appear to be quite marginal. This raises the question of whether the motivation addressed by the authors truly targets a critical or fundamental challenge in combinatorial optimization.

---

> ### Author Rebuttal · Authors · 2025-07-31
>
> We sincerely thank the reviewer for their valuable and detailed feedback. We are happy to provide below extensive answers to highlight the significance and originality of our approach.
>
>
> **W1 & Q1: about the formulation of multi-agent CO**
>
> We appreciate the opportunity to clarify these important concepts.
>
> In conventional CO, "agents" are indeed mathematical entities without the interactive properties we model. Our approach differs fundamentally in how we conceptualize and solve these problems.
>
> In our formulation, we model problems where multiple distinct entities must coordinate to construct a complete solution as multi-agent systems. Each "agent" corresponds to a physical entity (e.g., vehicle, machine, robot) that can independently take actions to build parts of the overall solution. This is formalized through our parallel multi-agent MDP in Section 4.1, where agents operate in a joint action space $A_1 \times ... × A_M$ rather than a single action space.
>
> The key distinctions are:
> - Multi-agent CO (PARCO): Multiple coordinating entities construct solution parts simultaneously with communication and conflict resolution.
> - Single-agent CO: One entity constructs the entire solution sequentially.
> - Conventional CO: Mathematical optimization without explicit agent modeling or coordination mechanisms.
>
> Regarding nodes, these represent the discrete decision points or locations in the problem space (customers in VRP, jobs in scheduling). The relationships between nodes are problem-specific (distances in routing, precedence in scheduling) and captured in our encoder representations and through masking in the decoder.
>
> Our decision variables are the actions taken by each agent at each time step, forming joint actions $\mathbf{a}_t = (a^1_t, ..., a^M_t)$. The optimization objective remains minimizing the original CO cost function, but achieved through coordinated parallel construction rather than centralized sequential planning.
>
> This formulation enables the communication layers and conflict resolution mechanisms that are central to our contribution, allowing agents to coordinate during solution construction rather than operating independently.
>
>
>
> **W2: About literature review**
>
>
> Thank you for raising this important point about our literature review and motivation. We appreciate your feedback and would be happy to expand our literature review in specific directions if you could suggest particular areas you would like us to focus on. Regarding the definition of multi-agent CO, we intentionally keep it broad because our approach is general-purpose and does not restrict to any single problem class; we view this generality as a key advantage of PARCO, allowing it to tackle diverse optimization scenarios with a unified framework.
>
> Your example of ride-hailing with polynomial-time algorithms like the KM algorithm is well-taken for certain task assignment problems. However, this does not apply to the significantly more complex problems we address. The combinatorial optimization problems we consider - particularly OMDCPDP and FFSP - are much harder computational challenges where traditional methods struggle considerably. Our experimental results demonstrate that PARCO not only finds substantially better solutions than existing approaches but does so more efficiently and with superior scalability. For instance, in FFSP, traditional solvers like Gurobi cannot even find feasible solutions for problems with $N > 20$ within reasonable time limits, while PARCO consistently outperforms all baselines across all tested scales. This highlights that while polynomial-time solutions may exist for some specific assignment problems, the broader class of multi-agent CO problems we target requires the sophisticated coordination and parallel decision-making capabilities that PARCO provides.
>
> **W3 & Q2: About the effectiveness of the conflict handler**
>
> Thank you for your detailed question about the conflict handler. We appreciate your concern about the fallback mechanism and would like to clarify several key points about its design and effectiveness.
>
> First, we want to emphasize that the fallback mechanism is only applied to actions that are in conflict and not selected as priority and not to every action. This selective application guarantees feasibility of construction while maintaining the parallel decision-making benefits. The priority-based selection uses learned probabilities from the model output (as shown in Fig. 3c, where "learned" priorities consistently outperform heuristic alternatives), making it an adaptive mechanism that leverages the neural network's learned representations rather than being purely fixed. The fallback actions (staying at current position for routing, keeping machine idle for scheduling) are domain-appropriate "do nothing" operations that maintain feasibility without disrupting solution quality, as agents can reconsider their choices in subsequent decoding steps.
>
> Regarding your question about more sophisticated conflict resolution methods, we acknowledge this as an interesting direction for future work. However, our empirical results demonstrate that the current approach already works quite well in practice: PARCO achieves state-of-the-art performance across all tested problems while maintaining computational efficiency. The simplicity of the fallback mechanism is actually a strength, as it provides reliable constraint satisfaction without adding computational overhead or training complexity. We chose this approach following the principle of using the simplest effective solution, similar to how masking is used in neural combinatorial optimization to guarantee feasible solution construction. The learned priority mechanism already provides the adaptive component you mentioned, while the fallback ensures robustness.
>
>
> **W4 & Q3: About lack of training curve plots**
>
> Thank you for pointing this out. While we cannot add figures or any other media due to the rebuttal policy, we will provide the below table with PARCO's convergence evaluated on the validation dataset  (training budgets refer to the percentage of the total training epochs):
>
> | Training budget  | 10% | 25% | 50% | 75% | 100% |
> |------|-----|-----|-----|-----|------|
> | HCVRP ($N=100$, $M=7$) | 5.13 ± 0.09 | 5.00 ± 0.06 | 4.90 ± 0.05 | 4.88 ± 0.03 | 4.79 ± 0.03 |
> | OMDCPDP ($N=100$, $M=20$) | 46.23 ± 0.21 | 45.57 ± 0.15 | 45.34 ± 0.11 | 44.87 ± 0.10 | 44.48 ± 0.09 |
> | FFSP ($N=100$, $M=12$) | 95.45 ± 0.52 | 94.32 ± 0.31 | 92.88 ± 0.20 | 92.15 ± 0.12 | 91.48 ± 0.08 |
>
> We can observe that PARCO is robust and yields consistent improvements.
>
> **Q4: What is the practical significance of the approach?**
>
> Thanks for asking this question, as it gives us an opportunity to further emphasize the strength of our approach. PARCO yields substantial imporvements over multiple problems, scales, and number of agents. In most problems we tested, it yields SOTA results even against heuristic approaches and a fraction of the cost.
>
> To further strengthen our results, we have tested PARCO for even larger scales in both number of nodes and number of agents during the rebuttal. We generated 16 new instances for $N=5000$ nodes and 3 different values of $M$ agents total of 48 instances of OMDCPDP and evaluate OR-Tools with 1 hour of runtime and greedy performance for HAM and PARCO. The results are in the table below:
>
>
>
> |        | $M=500$ |  |  | $M=750$ |  |  | $M=1,000$ |  |  |
> |--------|-------------|-------------|-------------|-------------|-------------|-------------|-------------|-------------|-------------|
> |        | Obj. | Gap | Time | Obj. | Gap | Time | Obj. | Gap | Time |
> | OR-Tools | 5575.73 | 134.06% | 3600s | 5127.46 | 115.24% | 3600s | 4974.81 | 188.10% | 3600s |
> | HAM | 4813.99 | 102.08% | 17.4s | 3732.06 |97.33%| 19.5s | 3258.26 | 88.69% | 22.3s |
> | PARCO | **2382.22** | **0.0%** | **0.21s** | **1891.28** | **0.0%** | **0.21s** | **1726.78** | **0.0%** | **0.22s** |
>
> As we can see from the results, PARCO excels at generalization in extremely large scales with $50\times$ the number of nodes and agents seen during training and up to a whopping $1,000$ agents. Thanks to its massively parallel structure, PARCO can solve such instances in a fraction of a second, with better results than OR-Tools -- at a $10,000\times$ speedup. This makes our PARCO ideal for real-world, real-time, large-scale complex applications.

---

> > ### Comment · Reviewer_NRcw · 2025-08-05
> >
> > Thank you for your detailed and comprehensive response. I would like to point out that, in my view, "task assignment" is not a trivial problem. Under the constraints where each driver can only take one order and each order can only be assigned to one driver, a simple fallback or "do nothing" conflict handling strategy may have significant drawbacks. Specifically, there may be multiple drivers competing for the same order, and if everyone "does nothing," it could result in no one taking the order, leading to a loss in overall revenue. Therefore, I still feel that the problem studied in this work might be too broadly defined, and the effectiveness of the proposed conflict handler may not be generally applicable.

---

> > > ### Author Response · Authors · 2025-08-05
> > >
> > > Thank you for your thoughtful follow-up and for giving us the opportunity to clarify this crucial point.
> > >
> > > Firstly, we sincerely apologize if our previous response appeared to imply that "task assignment" is a trivial problem; we agree with you that it is a complex and significant challenge. We will describe the below clarification tailored for this problem.
> > >
> > > There seems to have been a key misunderstanding of our conflict handling mechanism, which we would be grateful to correct. It is critical to distinguish between an action taken during a single step of the solution **construction process** and the **final, valid solution** that is generated.
> > >
> > > In the scenario you described -- where multiple drivers compete for the same order -- it is **not** the case that all of them default to a "do nothing" action that persists in the final solution. Instead, our Priority-based Conflict Handler (detailed in Algorithm 1) resolves this conflict dynamically during *construction*:
> > >
> > > 1.  When multiple agents select the same task, the handler identifies a single "winner" based on a priority -- e.g., the agent with the highest learned model confidence, as studied in Figure 3(c).
> > > 2.  This winning agent is assigned the task for that construction step.
> > > 3.  *Only the other competing agents* are assigned the "do nothing" fallback action. Crucially, this is a transient action within that **single solution construction step only**. It is a mechanism to defer their decision, not to finalize it.
> > > 4.  The construction process then continues. In the very next step, the deferred agents are free to select from the remaining pool of available tasks. This iterative process repeats until a **complete and feasible final solution** is constructed, in which **all tasks are assigned and all constraints are satisfied**.
> > >
> > > Therefore, the "do nothing" action is simply a temporary placeholder during the step-by-step construction for agents that do not have the highest priority, i.e., **the "all agents do nothing case" will never happen**, allowing for conflicts to be resolved sequentially while maintaining the efficiency of parallel decision-making. **The final solution generated by PARCO will be complete and valid**.
> > >
> > > Overall, our framework is designed for multi-agent Combinatorial Optimization (CO) problems that can be formulated as a Markov Decision Process (MDP), where a solution is constructed through a sequence of actions. The Priority-based Conflict Handler is a crucial component that enables this sequential process to be parallelized across multiple agents. Its design is therefore not overly broad, but rather **generally applicable to any multi-agent CO problem that fits this common sequential paradigm, including task assignment**.
> > >
> > > We hope this clarifies the distinction between the construction process and the final output, and resolves your concern about the Priority-based Conflict Handlers effectiveness. We remain available for further clarifications if needed.

---

> > > > ### Comment · Reviewer_NRcw · 2025-08-06
> > > >
> > > > Thank you for your clarifications, which have addressed many of my concerns. However, I still feel that the training stability of the proposed conflict handler is insufficient. In addition, I believe the writing could be further improved. For example, some of the terminology usage seems inconsistent with standard conventions. The phrase “multi-agent Combinatorial Optimization (CO) problems that can be formulated as a Markov Decision Process (MDP)” is somewhat confusing, as MDPs typically model single-agent sequential decision-making, while in multi-agent settings, frameworks such as Markov games or Dec-POMDPs are more appropriate. Mixing these terms can significantly mislead readers. That being said, I will raise my score accordingly.

---

> > > > > ### Author Response · Authors · 2025-08-06
> > > > >
> > > > > Thank you very much for your thoughtful engagement with our work and for raising your score! We sincerely appreciate your constructive final comments, which will help us further improve the quality of our paper.
> > > > >
> > > > > Regarding your concern about training stability, we will provide an analysis -- similar to the one provided to reviewer GvYn. Our gradient analysis demonstrates that:
> > > > >
> > > > > - The fallback ("do nothing") operation for agents that lose a conflict does not contribute to the policy gradient update.
> > > > >
> > > > > - The actions of "winning" agents are reinforced only based on the final reward of the complete solution, in line with the REINFORCE algorithm with POMO baseline.
> > > > >
> > > > > This mechanism ensures that agents are not incorrectly rewarded merely for winning a conflict if the overall solution is poor, thereby preventing overestimation bias and contributing directly to stable learning.
> > > > >
> > > > > Overall, we are confident in PARCO's stability, as evidenced by its successful, stable training and strong performance across three different and complex multi-agent CO domains (HCVRP, OMDCPDP, and FFSP).
> > > > >
> > > > >
> > > > > Regarding the writing and terminology, we again thank you for your sharp observation. We would like to provide more context for our initial framing. Prior work in this area has often modeled multi-agent CO problems using what is effectively a single-agent MDP formulation. In these sequential approaches, only one agent acts at a time until its entire task is complete, making the current agent an attribute of the state rather than a concurrent actor in a shared action space.
> > > > >
> > > > > Our work departs from this by formulating the problem as a multi-agent MDP, as outlined in Section 4.1. In our framework, all agents select actions in parallel at each timestep, which is a standard approach for multi-agent reinforcement learning and is functionally equivalent to a fully cooperative Markov Game, similarly to what is outlined by Boutilier [1]:
> > > > >
> > > > > > We think of MMDPs (Multi-agent Markov Decision Processes) as decision processes rather than games because of the existence of a joint utility function.
> > > > >
> > > > > Boutilier's Definition 1 aligns with our definition of multi-agent CO problems. We will add this reference and discussion in the camera-ready version of PARCO.
> > > > >
> > > > > Thank you once again for your expertise and for helping us strengthen our work and positioning. We hope these final changes will address your remaining concerns and strengthen your evaluation of PARCO's significance!
> > > > >
> > > > > As always, we remain available for further discussions.
> > > > >
> > > > > Best regards,
> > > > >
> > > > > PARCO's authors
> > > > >
> > > > > ---
> > > > >
> > > > > References
> > > > >
> > > > > [1] Boutilier, Craig. "Planning, learning and coordination in multiagent decision processes." TARK. Vol. 96. 1996.

---

### Official Review · Reviewer_HMqs · 2025-07-02

**Clarity:** 3
**Significance:** 3
**Originality:** 3
**Rating:** 4
**Confidence:** 3

**Summary:**

The authors propose a general reinforcement learning framework to handle multi-agent combinatorial optimization problems. The proposed framework mainly contains three components: (1) communication layers for multi-agent collaboration; (2) multi-pointer mechanism for low-latency multi-agent action generation; (3) priority-based conflict handler to deal with action conflicts based on learned priorities with allowance of fallback actions. Extensive experiments show that these designs effectively help the proposed approach achieve superior performance and efficiency in several challenging routing and scheduling problems.

**Questions:**

1. It is said in L196 that the communication layers are agent-count agnostic, but the self-attention layer in transformer may lead to fluctuated effect in multi-agent communication when the agent count becomes huge.
2. It seems like there are two stages of communication for multi-agent coordination, not only in the communication layers? The encoder module actually involves agent-agent interaction through self-attention MHA or even agent-node interactions through cross-attention MHA.

**Ethical Concerns:**

["NO or VERY MINOR ethics concerns only"]

**Limitations:**

Yes

**Quality:**

3

**Strengths And Weaknesses:**

Strengths:
1. The designs including the communication layers, multi-pointer mechanism, and the conflict handlers are compact in formulation and efficient to deal with multiple agents in parallel, clearly echoing the name "Parallel AutoRegressive Models".
2. The proposed communication layers are scalable in agent count, and it can generalized to out-of-distribution numbers of agents M according to the ablation results in table 2.
3. The whole framework is general to various combinatorial optimization problems including routing and scheduling tasks. It demonstrate extraordinary scalability and efficiency in solving time and speedups against related works, as shown in Table 1 and figure 4.

Weaknesses:
1. Lack of comparison with decentralized approaches in performance and efficiency, and any disadvantages against decentralized methods?
2. Lack of experiments comparing with related works modeling the multi-agent collaboration/communication other than MHA and MLP, for example, graph-based models?
3. Inconsistent depiction between figure and text body. As for the Multiple Pointer Mechanism, only context embeddings are fed in as key and value in Figure 2, but $\mathbf{h}_n$ plus dynamic node features are playing as keys and values in Eq. (5).

---

> ### Author Rebuttal · Authors · 2025-07-31
>
> We sincerely thank you for your time and for highlighting the strengths of our paper. We appreciate the insightful feedback, which has helped us further improve our work. Below, we address each of your concerns and questions.
>
>
> **W1 & W2: Comparison with Decentralized and Graph-Based Communication Methods**
>
> Thank you for this valuable suggestion. We agree that comparing PARCO with decentralized methods and alternative communication models like Graph Neural Networks (GNNs) provides a more complete picture. During the rebuttal period, we conducted new experiments on the min-max traveling salesman problem (mTSP), benchmarking PARCO against notable decentralized methods like GNN-DisPN [1], DAN [2], and GNN-based communication models like ScheduleNet [3].
>
> The results, shown in the table below (where g. stands for greedy and s. for sampling), demonstrate that PARCO consistently outperforms these methods in solution quality.
>
>
> | Method | N=50 |  |  | N=100 |  |  | N=200 |  |  | Gap(%) |
> |--------|------|------|------|-------|------|------|-------|------|------|--------|
> | **M** | **5** | **7** | **10** | **5** | **10** | **15** | **10** | **15** | **20** |  |
> | LKH3 (solver) | 2.00 | 1.95 | 1.91 | 2.20 | 1.97 | 1.98 | 2.04 | 2.00 | 1.97 | - |
> | OR-Tools (solver) | 2.04 | 1.96 | 1.96 | 2.36 | 2.29 | 2.25 | 2.57 | 2.59 | 2.59 | 14.42 |
> | GNN-DisPN (g.) | 2.14 | 2.10 | 1.99 | 2.56 | 2.22 | 2.04 | 2.97 | 2.30 | 2.15 | 13.45 |
> | DAN (g.) | 2.29 | 2.11 | 2.03 | 2.72 | 2.17 | 2.09 | 2.40 | 2.20 | 2.15 | 11.75 |
> | SchedNet (g.) | 2.17 | 2.07 | 1.98 | 2.59 | 2.13 | 2.07 | 2.45 | 2.24 | 2.17 | 10.16 |
> | **PARCO (g.)** | **2.12** | **2.00** | **1.92** | **2.47** | **2.02** | **1.98** | **2.28** | **2.06** | **1.99** | **4.43** |
> | DAN (*s.*) | 2.12 | 1.99 | 1.95 | 2.55 | 2.05 | 2.00 | 2.29 | 2.13 | 2.07 | 6.13 |
> | SchedNet (*s.*) | 2.07 | 1.99 | 1.92 | 2.43 | 2.03 | 1.99 | 2.25 | 2.08 | 2.05 | 4.29 |
> | **PARCO (s.)** | **2.07** | **1.98** | **1.91** | **2.38** | **1.99** | **1.98** | **2.22** | **2.03** | **1.98** | **2.80** |
>
>
> Centralized training with parallel decoding, as used in PARCO, offers distinct advantages by enabling global coordination and highly efficient solution construction. For example, GNN-based communication in models like ScheduleNet incurs significantly higher computational overhead, making our approach much faster.
>
> **W3: Clarification on Figure 2 and Dynamic Node Features**
>
> Thank you for pointing out this inconsistency. Figure 2 was simplified for visual clarity, but your interpretation of Equation (5) is correct. Dynamic node features ($\xi_t$) are indeed a part of our model. While these features are not used in the routing problems (where $\xi_t$ is zero), they are critical for the Flexible Flow Shop Problem (FFSP). In FFSP, $\xi_t$ encodes the time a job becomes available at a new stage, which is essential dynamic information for the machines (agents) to make correct scheduling decisions. We will update Figure 2 in the final version to fully reflect Equation (5).
>
>
> **Q1: Scalability**
>
> This is a valid concern. While self-attention complexity is quadratic with the number of agents, our experiments show that PARCO's performance remains robust and stable even at extreme scales. During the rebuttal, we stress-tested PARCO on the OMDCPDP with up to 5000 nodes and 1000 agents, 50 times larger than the training distributions.
>
>
> |        | $M=500$ |  |  | $M=750$ |  |  | $M=1,000$ |  |  |
> |--------|-------------|-------------|-------------|-------------|-------------|-------------|-------------|-------------|-------------|
> |        | Obj. | Gap | Time | Obj. | Gap | Time | Obj. | Gap | Time |
> | OR-Tools | 5575.73 | 134.06% | 3600s | 5127.46 | 115.24% | 3600s | 4974.81 | 188.10% | 3600s |
> | HAM | 4813.99 | 102.08% | 17.4s | 3732.06 |97.33%| 19.5s | 3258.26 | 88.69% | 22.3s |
> | PARCO | **2382.22** | **0.0%** | **0.21s** | **1891.28** | **0.0%** | **0.21s** | **1726.78** | **0.0%** | **0.22s** |
>
> As we can see from the results, PARCO excels at generalization in extremely large scales with $50\times$ the number of nodes and agents seen during training and up to a whopping $1,000$ agents. Thanks to its massively parallel structure, PARCO can solve such instances in a fraction of a second, with better results than OR-Tools at a $10,000\times$ speedup. This makes our PARCO ideal for real-world, real-time, large-scale complex applications.
>
>
> **Q2: Two Stages of Communication**
>
> This is an excellent observation. You are correct that one can view the model as having two stages of "communication":
> - Static Communication (Encoder): The encoder processes the initial agent and node features, establishing a baseline understanding of the problem structure and static agent-node relationships.
> - Dynamic Communication (Decoder): The Communication Layers provide a second, crucial stage of coordination that happens at every step of the decoding process. This allows agents to react to the dynamically changing state of the problem and each other's actions.
>
> Our ablation study in Figure 3a, which removes the explicit Communication Layers but keeps the same encoder, confirms that this second stage of dynamic collaboration is essential for achieving high-quality solutions.
>
> ---
>
> **References**
>
> [1] Hu, Yujiao, Yuan Yao, and Wee Sun Lee. "A reinforcement learning approach for optimizing multiple traveling salesman problems over graphs." Knowledge-Based Systems 204 (2020): 106244.
>
> [2] Cao, Yuhong, Zhanhong Sun, and Guillaume Sartoretti. "Dan: Decentralized attention-based neural network for the minmax multiple traveling salesman problem." International Symposium on Distributed Autonomous Robotic Systems. Cham: Springer Nature Switzerland, 2022.
>
> [3] Park, Junyoung, Changhyun Kwon, and Jinkyoo Park. "Learn to Solve the Min-Max Multiple Traveling Salesmen Problem with Reinforcement Learning." AAMAS. Vol. 22. 2023.

---

### Official Review · Reviewer_ycp1 · 2025-07-03

**Clarity:** 3
**Significance:** 2
**Originality:** 2
**Rating:** 5
**Confidence:** 1

**Summary:**

PARCO is a reinforcement learning framework designed to efficiently solve complex multi-agent combinatorial optimization problems. It introduces three key innovations: (1) a transformer-based communication layer for agent coordination, (2) a multiple-pointer mechanism for parallel decision-making, and (3) a priority-based conflict resolution system. Evaluated on tasks like multi-agent vehicle routing and scheduling, PARCO outperforms state-of-the-art learning-based methods in both solution quality and generalization.

**Questions:**

No particular question, I think this is a well written paper and can be accepted.

**Ethical Concerns:**

["NO or VERY MINOR ethics concerns only"]

**Limitations:**

yes

**Quality:**

3

**Strengths And Weaknesses:**

Strengths:
1. the empirical improvement is clear, including the performance and the efficiency.
2. the architecture design seems reasonable, including multi-agent encoder, communication layer, and multi-pointer decode.

Weakness:
1. no particular weakness, probably lack of guarantee on optimality.

---

> ### Author Rebuttal · Authors · 2025-07-31
>
> We sincerely thank you for your time and for providing a positive and encouraging review, acknowledging the design choices of our PARCO method, our empirical results, as well as the clear writing. We will answer your concern below:
>
>
> **W1: About guarantee on optimality**
>
> Regarding the lack of an optimality guarantee, we agree with the reviewer's observation. This is a common and accepted trade-off for heuristic and learning-based methods designed to tackle NP-hard problems. Our primary objective is to produce high-quality solutions in the shortest possible time, which is a critical requirement for many real-world applications where exact methods are computationally infeasible. In this sense, PARCO can produce a high-quality solution at a fraction of the cost of traditional methods while outperforming all learning-based ones. This is validated across numerous benchmarks and ablation studies in the paper. While PARCO does not offer an optimality guarantee, it successfully provides a state-of-the-art, scalable, and efficient solver for practical use.

---

### Note · Authors · 2025-08-13

Dear Reviewers, Area Chairs, and Program Committee,

We sincerely thank you for your time, insightful feedback, and constructive discussions during the review process. Your comments have been invaluable in clarifying and strengthening our work.

The rebuttal period allowed us to substantiate further PARCO's contribution: a novel parallel autoregressive framework capable of efficiently solving diverse and complex multi-agent combinatorial optimization problems (HCVRP, OMDCPDP, FFSP, and our newly added mTSP benchmark).

Through the discussions, we addressed key design aspects. We demonstrated that the Priority-based Conflict Handler is both **adaptive** -- via learned priorities -- and **stable** -- by preventing overestimation bias through zero-gradient fallback actions as demonstrated in our new analysis and training convergence results -- directly responding to points from Reviewers GvYn and NRcw. We will also refine our terminology, as suggested by Reviewer NRcw, to better align with cooperative multi-agent MDP/Markov Game literature (e.g., Boutilier), improving clarity for the community.

To address the question of **practical significance**, we conducted new **very-large-scale experiments** (up to 5000 nodes and 1000 agents -- 50x more than the training distribution). Results show PARCO delivers superior solutions over **10,000× faster** than strong traditional solvers, underscoring its value for **scalable, real-time applications**.

We are confident in PARCO's contribution and potential impact, and we look forward to incorporating your feedback into the camera-ready version. Thank you again for your careful consideration.

Best regards,

The PARCO Authors

---

### Decision · Program_Chairs · 2025-09-17

**Decision:**

Accept (poster)

**Comment:**

Summary:

This paper introduces PARCO (Parallel AutoRegressive Combinatorial Optimization), a novel reinforcement learning framework designed to solve multi-agent combinatorial optimization (MACO) problems. The core contribution is a parallel solution construction method that addresses the high latency and poor coordination of traditional sequential autoregressive models. PARCO integrates three key components: (1) transformer-based Communication Layers to facilitate effective coordination among agents during decoding; (2) a Multiple Pointer Mechanism that enables simultaneous, low-latency decision-making for all agents; and (3) a Priority-based Conflict Handler that uses learned priorities to resolve action conflicts and ensure solution feasibility. The authors evaluate PARCO on several challenging multi-agent routing (HCVRP, OMDCPDP) and scheduling (FFSP) problems, demonstrating that their approach outperforms state-of-the-art learning-based methods and traditional solvers in terms of solution quality, computational efficiency, and generalization.

Strengths:
- Strong Empirical Performance and Efficiency: PARCO consistently achieves state-of-the-art results, outperforming strong learning-based and traditional baselines (e.g., OR-Tools) across multiple complex MACO domains. Its parallel construction method leads to remarkable computational speedups, making it suitable for real-time applications.
- Novel and Sound Methodology: The proposed parallel autoregressive framework is a significant and well-motivated departure from prior sequential methods. The architectural components, particularly the Communication Layers and Multiple Pointer Mechanism, are considered well-designed and theoretically sound for enhancing agent coordination and efficiency.
- Excellent Generalization and Scalability: The framework demonstrates impressive zero-shot generalization to problem instances with significantly more nodes and agents (up to 50x larger than seen in training). The authors substantiated this during the rebuttal with new large-scale experiments, showcasing robust performance with up to 1,000 agents.

Weaknesses:
- Clarity and Terminology: The most consistent remaining critique, raised by Reviewer NRcw, pertains to the clarity of the writing and the use of non-standard terminology. Specifically, framing the multi-agent problem as a Markov Decision Process (MDP) was noted as potentially confusing, with "Markov Game" being the more appropriate term from literature. The authors have acknowledged this and committed to revising the text for the camera-ready version.

The Most Important Reason for Decision:

The decision to recommend acceptance is based on the fact that the paper's significant strengths and contributions convincingly outweigh its remaining, and largely addressable, weaknesses. The authors propose a novel, efficient, and high-performing framework for a challenging and practical class of problems. The quality and thoroughness of the author rebuttal were exceptional; they provided substantial new experiments (including a new mTSP benchmark and very large-scale tests) that successfully addressed the vast majority of the initial, more critical concerns raised by the reviewers. The paper makes a solid contribution to the field of neural combinatorial optimization.

Summary of Rebuttal and Changes

The discussion period was highly productive and pivotal. Initially, the paper received mixed reviews, with significant concerns from Reviewer NRcw. Key points raised and addressed were:

Concern 1: Lack of Comparison with Other Methods

Points Raised: Reviewer HMqs noted that the paper lacked a comparison with decentralized approaches and alternative communication models, such as those based on Graph Neural Networks (GNNs).

Authors' Response: During the rebuttal, the authors conducted new experiments on the min-max traveling salesman problem (mTSP). They benchmarked PARCO against notable decentralized methods (GNN-DisPN, DAN) and a GNN-based communication model (ScheduleNet), with results showing PARCO consistently outperforming these baselines in solution quality.

Final Decision Weight: This was a significant addition that addressed a clear gap in the original submission. By providing new, targeted experiments, the authors substantially strengthened the paper's empirical claims and demonstrated the superiority of their approach. This weighed heavily in favor of acceptance.

Concern 2: Effectiveness and Design of the Conflict Handler

Points Raised: Reviewer NRcw raised major concerns about the conflict handler, describing it as a "fixed, non-adaptive, and non-learnable fallback mechanism". Reviewer GvYn also questioned whether using the policy's output for priority could introduce overestimation bias.

Authors' Response: The authors provided a detailed clarification, explaining that the mechanism is adaptive because it uses learned priorities from the model's output to select a single "winner" in a conflict. To address the bias concern, they provided a gradient-based analysis showing that the deterministic fallback has a zero gradient, thus preventing reinforcement and overestimation.

Final Decision Weight: This was a critical point of discussion. The authors' thorough response corrected a key misunderstanding of their method and successfully addressed the concerns about adaptiveness and bias. This was significant in mitigating what was initially a significant reason for potential rejection.

Concern 3: Lack of Training Analysis and Perceived Marginal Improvement

Points Raised: Reviewers NRcw and GvYn noted the absence of training curves to evaluate convergence and stability. Reviewer NRcw also initially felt the reported performance improvements were "quite marginal".

Authors' Response: The authors provided tables with convergence data and committed to including full training curve plots in the final version. To counter the claim of marginal improvement and demonstrate practical significance, they conducted new, very-large-scale experiments (up to 5000 nodes and 1000 agents), showing that PARCO finds superior solutions up to 10,000x faster than strong traditional solvers like OR-Tools.

Final Decision Weight: The new large-scale results were particularly compelling, highlighting the method's strengths in scalability and efficiency, which significantly boosted the paper's perceived impact and weighed strongly in favor of acceptance.

Concern 4: Problem Formulation and Terminology

Points Raised: Reviewer NRcw found the terminology inconsistent with standard multi-agent literature, specifically the use of "Markov Decision Process (MDP)" for a multi-agent setting, which could be confusing for readers.

Authors' Response: The authors acknowledged the reviewer's point, explaining that their formulation is functionally equivalent to a fully cooperative Markov Game. They agreed to add a reference to relevant literature (e.g., Boutilier) and clarify their terminology in the camera-ready version to better align with standard conventions.

Final Decision Weight: While a presentation issue, resolving it was important for ensuring the paper's correctness and accessibility. This cooperative engagement was viewed positively in the final recommendation.